# Self-administered mindfulness interventions reduce stress in a large, randomized controlled multi-site study

Mindfulness witnessed a substantial popularity surge in the past decade, especially as digitally self-administered interventions became available at relatively low costs. Yet, it is uncertain whether they effectively help reduce stress. In a preregistered (OSF https://doi.org/10.17605/OSF.IO/UF4JZ; retrospective registration at ClinicalTrials.gov NCT06308744) multi-site study ($n_{sites}$ = 37, $n_{participants}$ = 2,239, 70.4% women, $M_{age}$ = 22.4, s.d.$_{age}$ = 10.1, all fluent English speakers), we experimentally tested whether four single, standalone mindfulness exercises effectively reduced stress, using Bayesian mixed-effects models. All exercises proved to be more efficacious than the active control. We observed a mean difference of 0.27 ($d$ = −0.56; 95% confidence interval, −0.43 to −0.69) between the control condition ($M$ = 1.95, s.d. = 0.50) and the condition with the largest stress reduction (body scan: $M$ = 1.68, s.d. = 0.46). Our findings suggest that mindfulness may be beneficial for reducing self-reported short-term stress for English speakers from higher-income countries.

Mindfulness meditation is defined as 'paying attention in a particular way: on purpose, in the present moment and nonjudgmentally'[1]. It thus emphasizes attention to the present moment, with awareness of one's bodily sensations or one's mental content such as thoughts, emotions and memories. Engaging in mindfulness meditation appears simple: one is asked to focus one's attention on the breath and on the present moment, without needing complex postures, settings or apparatus. Partly because of this apparent simplicity, mindfulness meditation protocols that can be self-administered (often referred to as self-help mindfulness interventions) have increased in accessibility and popularity in recent years[2]. Their appeal relies on costs lower than for those administered by professionals, such as mindfulness-based stress reduction (MBSR) programmes[3], and on easier accessibility[4–6] owing to diverse formats (for example, self-help books, computer programmes, smartphone apps and audio and video recordings).

Notwithstanding their popularity, access to such mindfulness tools remains restricted to those who can afford both the costs and the time necessary to practice. Yet, despite having millions of users, evidence for the effectiveness of these mindfulness interventions is debated and at least two key empirical questions remain unanswered. First, are these types of interventions truly effective in reducing stress levels? And second, which self-administered mindfulness exercises, from the plethora of those available, might work best? We attempted to answer these questions first by conducting a survey among mindfulness practitioners to identify the mindfulness exercises that are most likely to reduce stress. On the basis of the results of the survey, we then designed a multi-site, highly powered study to test the effects and the boundary conditions of four self-administered mindfulness meditation exercises on stress reduction.

Compared to established mindfulness protocols (for example, MBSR[3]), self-administered mindfulness exercises present fewer constraints. They do not require the physical presence of an instructor because they include prerecorded protocols and they allow practitioners to meditate at the time and place of their choosing[6]. And while some established protocols need individuals to sustain practice for at least 8 weeks, many self-administered mindfulness interventions hold promises for reducing stress levels despite being short and allowing one to practice if and when one decides. It is thus important

✉e-mail: Alessandro_Sparacio@sics.a-star.edu.sg; hans@absl.io; ivan.ropovik@gmail.com; G.Jiga@swansea.ac.uk

**Table 1 | Means and s.d. of self-reported stress levels of the four Bayesian mixed-effects models with the active control for the STAI Form Y-1**

| Condition | $n$ | $M$ | s.d. | $BF_{10}$ |
|---|---|---|---|---|
| Active control condition | 478 | 1.95 | 0.50 | – |
| Body scan | 449 | 1.68 | 0.46 | $3.7 \times 10^{11}$ |
| Mindful breathing | 469 | 1.73 | 0.50 | $2.3 \times 10^{5}$ |
| Loving kindness | 427 | 1.70 | 0.49 | $1.1 \times 10^{7}$ |
| Mindful walking | 416 | 1.73 | 0.46 | $4.8 \times 10^{2}$ |

A positive Bayes factor ($BF_{10}$) denotes increasing evidence of $H_1$ compared to $H_0$.

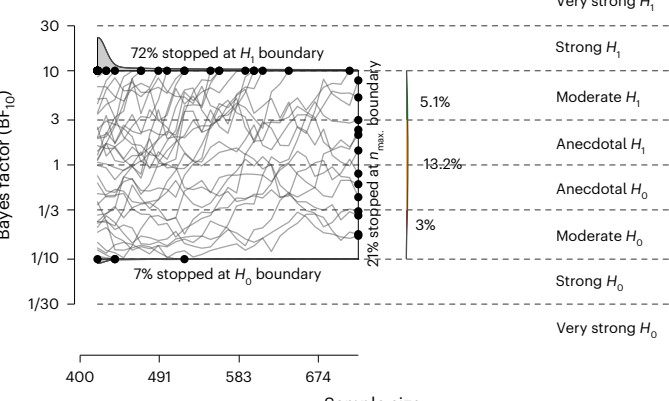

**Fig. 1 | Simulation of the Bayesian two-sided sequential design.** After 10,000 iterations, the simulation indicates that under the proposed design, there is a 79% chance (72% under $H_1$ and 7% erroneously under $H_0$) that the test will reach compelling evidence boundaries ($BF_{10} = 10$ or 1/10). There is a 21% chance that the test will conclude by reaching the maximum (max.) sample size of 720 per condition, with a 5% probability of providing some evidence in favour of $H_1$ ($BF_{10} > 3$).

to understand whether they indeed bring about the expected results. While some studies[2,7] and a recent meta-analysis[8] have shown reductions in self-reported stress following self-administered mindfulness, others[9] did not find evidence that such training effectively decreased perceived stress and a meta-analysis failed to find robust effects in this direction after accounting for publication bias[10].

A different, albeit important issue is that many such exercises have been empirically examined as part of longer sequences that include more than one exercise, making it difficult to conclude what specific effect each exercise can have on reducing stress. Some studies have tested single brief mindfulness exercises[11,12], however, to our knowledge, none investigated the effectivess of brief standalone mindfulness exercises on stress reduction. Others[13] divided the plethora of mindfulness exercises into three categories reflective of their focus, namely 'awareness', 'present experience' and 'acceptance'. Awareness mindfulness exercises typically involve a sequence of steps going from disengaging from an automatic train of thought (for example, interrupting repetitive thoughts by taking a long breath) to focusing the attention on an object that is used as an 'anchor' (for example, the breath and body parts), returning the attention to the object of focus when one realizes they had been distracted and watching where the mind wanders next. Present experience mindfulness exercises instruct participants to pay attention completely to the activity being carried out (for example, bringing the attention to the sole of the foot while walking). If the mind wanders, the instructions given aim to help the practitioner redirect their attention to the present moment. Acceptance mindfulness exercises are characterized by applying a non-judgmental attitude of kindness and curiosity to one's experience. Practitioners are invited to cultivate positive feelings towards themselves and others (for example, directing loving kindness to themselves or to someone else). While these different categories may share some common features, for the purposes of the present investigation we maintained this system of classification because it allowed us to better understand the potential applied value of such self-administered mindfulness exercises.

Finally, the potential moderating influence of different personality traits on the effects of these exercises remains largely unexplored. Previous research has indicated that neuroticism may moderate the psychological effects of mindfulness training[14,15]. A meta-analysis appraising the evidence of 29 studies found that neuroticism exhibits the most pronounced association with self-reported individual differences in mindfulness among the Big Five personality traits ($r = -0.45$; ref. 16). Furthermore, one study found that individuals who scored higher in neuroticism showed a more significant decrease in psychological distress and improvement of overall wellbeing when compared to a control group after participating in an MBSR. While this study suggested that neuroticism moderated the effect, the power of the design (with $n = 244$) to detect smaller but still theoretically meaningful interaction was modest[17] and the authors acknowledged that the use of four possible moderators for each outcome may have inflated type 1 errors[14].

**Table 2 | Effect sizes for each mindfulness condition tested against the active control, along with their 95% confidence intervals (CIs) and standard errors of the estimate (s.e.)**

| Condition test (against control) | Cohens' $d$ [95% CI] | s.e. |
|---|---|---|
| Body scan | −0.56 [−0.43, −0.69] | 0.07 |
| Mindful breathing | −0.46 [−0.30, −0.61] | 0.08 |
| Loving kindness | −0.48 [−0.35, −0.62] | 0.07 |
| Mindful walking | −0.45 [−0.32, −0.59] | 0.07 |

Therefore, the primary objective of this multi-site project was to test the comparative effectiveness of self-administered mindfulness exercises in reducing individuals' stress levels when compared to a non-mindful active control condition. We proposed that participants allocated to any experimental (mindfulness) condition would experience lower self-reported stress levels compared to participants allocated to an active control condition. The secondary objective was to explore whether these effects are moderated by participants' levels of neuroticism and by their English language proficiency. To justify the latter factor's potential moderating role, we looked at how language plays a role in the acquisition of knowledge to make meaning of emotional experiences and perceptions[18]. If certain levels of knowledge of a particular language are not reached, the processes of making meaning out of emotional experiences could be compromised.

## Results

### Confirmatory analyses of mindfulness versus control effect

We recoded the reverse items and then averaged the scores for the 20 state-focused items of the state-trait anxiety inventory (STAI) Form Y-1 (ref. 19), the self-reported measure of stress. The experimental condition with the highest Bayes factor was the body scan, with a Bayes factor of $3.69 \times 10^{11}$, indicating that the observed data are $3.69 \times 10^{11}$ times more likely to occur under $H_1$ (that is, participants report lower self-reported levels of stress in the mindfulness conditions compared to the control condition) than under $H_0$ (that is, there is no difference between conditions in self-reported stress levels), thus denoting 'extreme evidence'[20]. This confirms the hypothesis that the body scan meditation exercise reduced self-reported stress compared to the active control condition (Table 1). All other mindfulness conditions

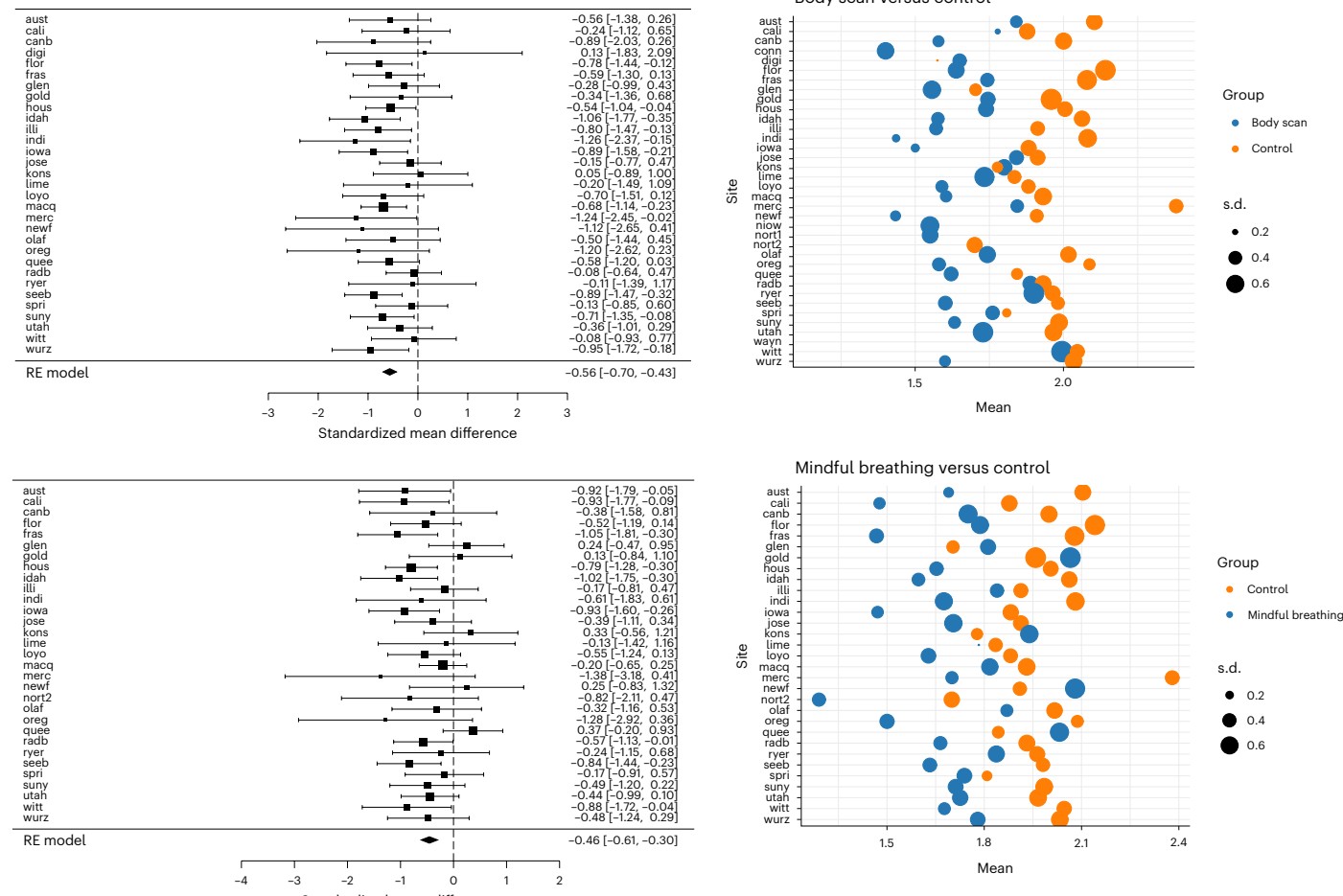

**Fig. 2 | Forest plot and bubble plot for body scan and mindful breathing.** On the left are the Forest plots for the effects of body scan (upper one) and mindful breathing (lower one) versus control, using Cohen's *d* as the effect size measure. Black boxes represent site-level effect size estimation of the random-effects (RE) model and the horizontal lines represent the associated CIs. The diamond represents overall effect size estimate and the 95% CI (*n* = 2,239). On the right are the bubble plots showing site-level means and s.d. The list of sites and abbreviations can be found here: https://osf.io/bdwu8.

also surpassed the threshold of compelling evidence of 10 in favour of $H_1$ compared to the active control (Table 1). The Bayesian mixed-effects models provided strong evidence that all four mindfulness conditions were effective in reducing participants' self-reported stress levels compared to the active control condition (Fig. 1 gives a simulation of the Bayes factor design).

**Exploratory analyses**

**Cohen's *d* for each condition compared to the active control condition.** We calculated Cohen's *d* for each condition test using the escalc function of the metafor package using sample means (*M*) and sample s.d. Even if we relied on a Bayesian framework, we used Cohen's *d* as an estimate of the magnitude of the effect because Cohen's *d* can be interpreted as the standard mean difference between two independent samples. Table 2 summarizes the effect sizes for all the conditions when compared to the control condition.

**Heterogeneity per site.** For each of the mindfulness exercises, we did not detect significant heterogeneity. Forest plot (right side) plotted means and s.d. for self-reported levels of stress for each mindfulness condition exercise compared to the active control condition (left side) are shown in Figs. 2 and 3. Additionally, in Table 3 we reported the heterogeneity values for each mindfulness condition across sites.

**Emotion dimensions.** We explored the effects of the mindfulness exercises on the dimensions of pleasure, arousal and dominance as compared to the active control condition again using four Bayesian mixed-effects models. We found that only for the dimension of pleasure and only for the mindful breathing condition the Bayes factor favoured $H_1$, surpassing the set threshold ($BF_{10} = 16.1$), indicating that participants who engaged in mindful breathing felt more pleasant than participants who listened to the story in the active control condition.

**Moderation by neuroticism.** We investigated whether neuroticism moderated the relationship between mindfulness exercises and stress. We merged the four mindfulness conditions and compared the merged conditions to the active control condition to achieve higher power. We failed to find any evidence for the moderation of neuroticism as the ratio between the two models (the full model and the one with only the interaction) yielded an inconclusive Bayes factor ($BF_{10} = 0.11$).

**Moderation by English language proficiency.** We investigated whether participants' English language proficiency moderated the effect of the mindfulness exercises on stress. Of the total 2,239 participants included in the analyses, 647 were non-native English speakers at least C1/C2 level, while 1,592 were native English speakers. We again merged the mindfulness conditions into a single group to increase

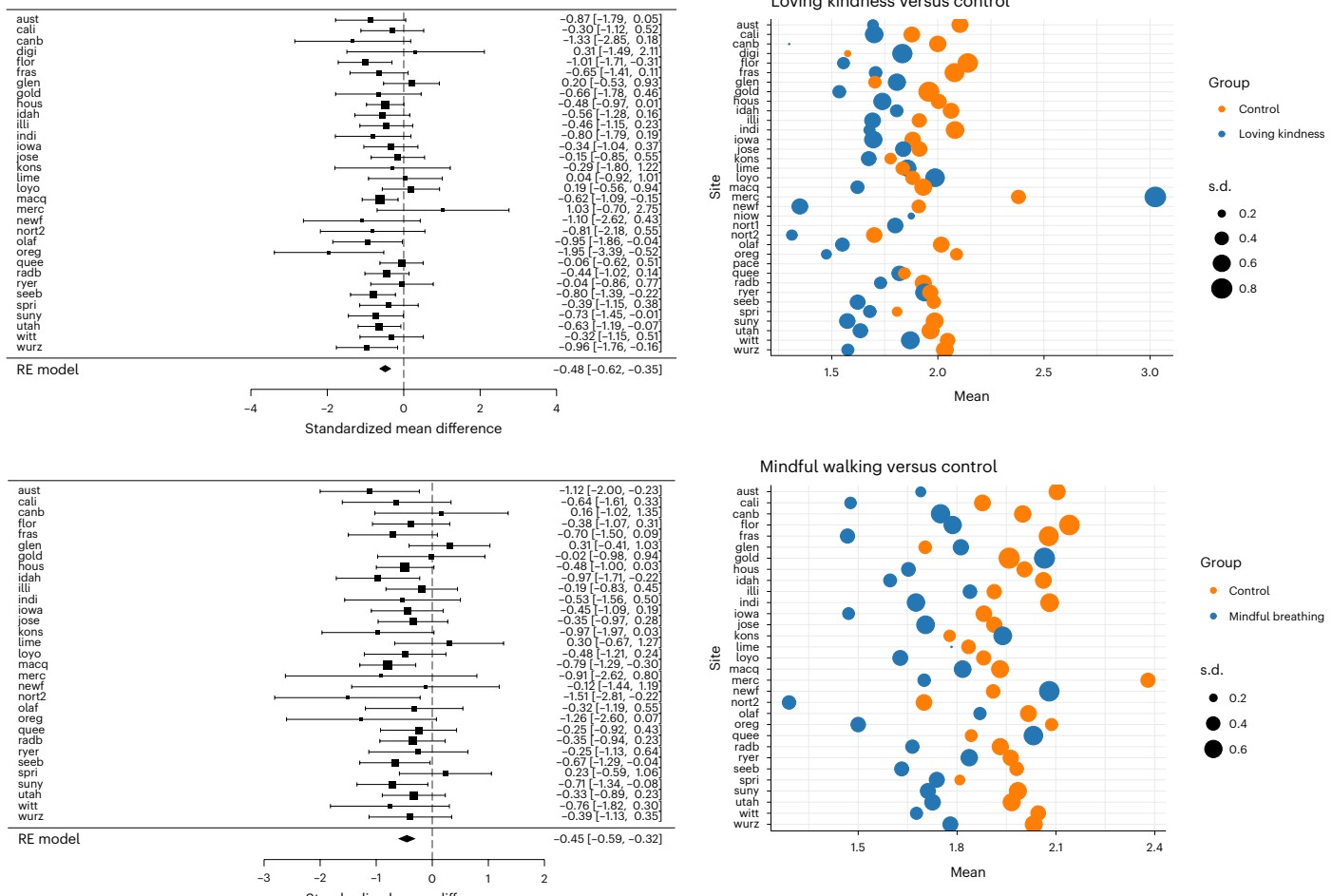

**Fig. 3 | Forest plot and bubble plot for loving kindness and mindful walking.** On the left are the Forest plots for the effects of loving kindness (upper one) and mindful walking (lower one) versus control, using Cohen's *d* as the effect size measure. Black boxes represent site-level effect size estimation of the RE model and the horizontal lines represent the associated CIs. The diamond represents overall effect size estimate and the 95% CI (*n* = 2,239). On the right are the bubble plots showing site-level means and s.d. The list of sites and abbreviations can be found here: https://osf.io/bdwu8.

statistical power. We did not find evidence for an interaction effect between mindfulness conditions and participants' English language proficiency (BF$_{10}$ = 0.05).

### Robustness analyses

We examined whether the difference in stress levels between the experimental (mindfulness) and the control condition could be due to one particular story excerpt ('Silverview' by John le Carré[21]) being perceived as more anxiogenic than the others (Table 4). We conducted three independent *t*-tests and found a slight discrepancy in self-reported stress levels between participants who listened to John le Carré's 'Silverview' excerpt and those who listened to Tolkien's 'Smith of Wootton Major' excerpt[22], *t*(313.75) = 2.71, *P* = 0.007. To address the issue of several comparisons, we applied the Bonferroni correction, yielding an adjusted *P* = 0.021. Notably, even after applying the Bonferroni correction, a statistically significant difference persisted between the two excerpts. These results indicate that participants exposed to the 'Silverview' excerpt experienced higher levels of stress compared to those exposed to the 'Smith of Wootton Major' excerpt. To test whether this may have affected the overall results, we re-ran the main analyses excluding participants who listened to 'Silverview' (*n* = 157). This participant exclusion resulted in a significant decrease in power because the control group was reduced from 478 to 321 participants; nevertheless, the Bayes factor remained above the threshold for compelling

evidence in three of the mindfulness conditions (body scan, mindful breathing and loving kindness) but not in the mindful walking condition (BF$_{10}$ = 0.08; Table 5).

### Discussion

We investigated whether four different mindfulness exercises were independently effective in reducing participants' stress levels as compared to the active control condition. We found that all four mindfulness exercises (body scan, mindful breathing, mindful walking and loving kindness) decreased participants' self-reported stress compared to listening to one of the three story excerpts that was part of the active control condition.

The current research aimed to fill a knowledge gap regarding the efficacy of brief, self-administered mindfulness interventions for reducing stress. Recent meta-analyses either failed to find evidence in favour of such effects[9,10] or detected them, albeit small in magnitude[2,8], potentially because of the high risk of bias or small sample of the studies included and insufficient power[23]. Other such tests included solely a passive, rather than an active, control condition[7], while still others did not adhere to open science practices by lacking preregistration[24,25].

The present multi-site design attempted to provide solutions for these shortcomings. Indeed, compared with previous studies, the present multi-site design was adequately powered, compared each mindfulness condition with an active control group (and not a

**Table 3 | Heterogeneity values for each mindfulness condition across sites**

| Condition | Cochran's Q-test (P value) | τ | $I^2$ |
|---|---|---|---|
| Body scan | 0.83 | 0 | 0% |
| Mindful breathing | 0.17 | 0.21 | 24.24% |
| Loving kindness | 0.50 | 0 | 0% |
| Mindful walking | 0.67 | 0 | 0% |

τ, s.d. of the distribution of true effects; $I^2$, proportion of total variation in study estimates due to heterogeneity.

**Table 4 | Means and s.d. of scores on the STAI Form Y-1 for each story of the control condition**

| Story excerpt | Length (min) | Word count | M | s.d. |
|---|---|---|---|---|
| 'Silverview' by John le Carré | 15.01 | 1,838 | 2.04 | 0.51 |
| 'The Old Man and the Sea' by Ernest Hemingway | 14.19 | 2,039 | 1.93 | 0.47 |
| 'Smith of Wootton Major' by J. R. R. Tolkien | 14.58 | 2,309 | 1.88 | 0.50 |

**Table 5 | Results of the four independent comparisons with the active control for the STAI Form Y-1, after excluding participants who listened to the 'Silverview' excerpt**

| Condition | n | M | s.d. | $BF_{10}$ |
|---|---|---|---|---|
| Active control condition | 321 | 1.91 | 0.49 | – |
| Body scan | 449 | 1.68 | 0.46 | $2.3×10^5$ |
| Mindful breathing | 469 | 1.73 | 0.50 | 16.65 |
| Loving kindness | 427 | 1.70 | 0.49 | 118.10 |
| Mindful walking | 416 | 1.73 | 0.46 | 0.08 |

passive control group or a waiting list) and was preregistered. The results can thus serve as a reliable basis for building testing protocols of self-administered mindfulness effects because they suggest that the four mindfulness exercises included in the study are slightly effective in reducing stress levels.

This project is an important step toward obtaining high-powered tests of the efficacy of self-administered mindfulness exercises for reducing stress. On the one hand, the current multi-site study showcases how even short mindfulness exercises can be valuable tools in situations when short-term mood regulation is necessary, such as withstanding a stressful exam or calming oneself in a road-rage situation[26]. The possibility that short-term mindfulness practice adds to one's repertoire of skills to reduce stress need not harm nor challenge the popular expectation that mindfulness meditation brings about positive results only via prolonged practice. Learning to practice mindfulness in a shorter time than traditional protocols typically require is a valuable asset for people for whom longer time commitment for mindfulness is a capacity- or motivation-based deterrent[27].

Understanding the optimal timing to learn mindfulness skills or the conditions in which mindfulness induces effects which are longer-term compared to those observed in the present experiment are important questions, yet they extend beyond the scope of the present research. Notwithstanding the absence of high-powered, preregistered studies which would make for a more reliable body of knowledge on these topics, some existent data yet allow partial answers. In line with the extended model of emotion regulation[28], mindfulness skills mastered before a stressful situation occurs can allow someone extra flexibility to regulate antecedents of emotional reactions, such as which aspects one pays attention to (attentional deployment) or the way one cognitively represents the stressful situation (cognitive change). For example, an 8-week randomized controlled trial of mindfulness completed in the year leading to the examination period significantly reduced students' psychological distress during that same examination period[29]. Longer, for example, 8-week mindfulness protocols such as MBSR can enhance trait/dispositional mindfulness (the inherent capacity to be in the present moment[15,30]) and people's mindfulness self-efficacy (one's perceived ability to maintain non-judgemental awareness in different situations). Therefore, for individuals who already possess high levels of trait mindfulness, the

timing of mindfulness exercises may be less crucial, as they already exhibit a disposition that helps reduce their susceptibility to stressors. Nevertheless, more preregistered, high-powered studies need to be conducted on the topic to conclusively determine the ideal timing for mindfulness exercises and their potential for long-term changes.

Despite the strengths of the current multi-site project, some limitations must be considered. The effects of each mindfulness exercise on stress were rather small and relied on self-reported stress. Such assessments may limit the validity of the present findings. Participants may lack introspective ability leading to biased estimates about their levels of stress[31] and may be subjected to demand characteristics effects[32,33]. Future research using physiological assessments of the autonomic nervous system (for example, assessment of catecholamines, assessment of the autonomic nervous system via skin conductance, cortisol, heart rate and systolic and diastolic blood pressure[34,35]) may help limit such problems. Thus, future studies investigating the efficacy of single brief self-administered mindfulness exercises should include both psychological and physiological measures to render more reliable estimates of stress levels and to rule out the possibility of a demand characteristics effect. Another potential limitation is the choice of control condition in our study. We found that participants who listened to the excerpt from 'Silverview'[21] by John le Carré exhibited higher levels of state anxiety compared to participants who listened to the two other story excerpts. We conducted a sensitivity analysis by excluding the former and found that only three mindfulness conditions (body scan, loving kindness and mindful breathing) led to a significant reduction in self-reported stress when compared to the control condition, which now involved listening to only two different randomly sampled stories. However, we did not observe a similar significant effect for the mindful walking group. This outcome may be attributed to a reduced statistical power in the control group which in this analysis loses one-third of the participants (decreasing from 478 to 321). Finally, we believe that it is important to consider several limitations on the generalizability of the results of this study[36]: Our findings only apply directly to participants who are (1) older than 18, (2) fluent/native English speakers living in Australia, Europe, the United Kingdom, Canada and the United States, (3) non-meditators, (4) do not have a history of mental illness and (5) mostly students (94.2%). Further research is needed to test whether the findings of the present study will indeed generalize to other populations.

In conclusion, we have conducted a large-scale project investigating the efficacy of single brief mindfulness interventions in a multi-site study conducted over 37 sites and including 2,239 valid observations. The limitations of the study notwithstanding, we found that each of the four mindfulness exercises (body scan, mindful breathing, mindful walking and loving kindness) was slightly more efficacious in reducing self-reported stress as compared to the active control condition. These interventions should be intended as being effective in the short-term and are unlikely to affect dispositional traits (such as chronic stress). Although we found an effect for single brief mindfulness exercises, our multi-site study carries the limitations of using only self-report measures. Well-powered studies with a physiological assessment of

the autonomic nervous system are thus necessary to corroborate the results of the current multi-site project.

## Methods

### Ethical regulations statement

This research project complied with all ethical regulations for research involving human participants laid out by the host organization, Swansea University. Approval was granted by the School of Psychology's Research Ethics Committee. The participating sites either received ethical approval from their local institutional review boards (IRBs) or stated that they were exempt. Swansea University and Université Grenoble Alpes carried out the administrative organization for the study. Swansea University was also the data controller for this project. Informed consent was obtained from all participants before collecting any data. Participants' personal data were processed for the purposes outlined in the information sheet. The project was conducted in line with the CO-RE Lab Lab Philosophy v.5 (ref. 37). The current multi-site project (ClinicalTrials.gov: NCT06308744) followed the route of a parallel randomized controlled trial. All materials used in the study, including the preregistered document (https://osf.io/us5ae), the ethics (IRB) approval documents of all the sites involved in the project and the meditation scripts are available on our Open Science Framework (OSF) page (https://osf.io/6w2zm/) and in our ClinicalTrials.gov registration. The data analytic script can be found on the GitHub repository of the project (https://github.com/alessandro992/A-large-multisite-test-of-self-administered-mindfulness) and on the OSF page (https://osf.io/6w2zm/).

### Participants

Data were collected between 23 March and 30 June 2022. We limited participation in the study to English native speakers or participants who self-assessed their English language proficiency at the C1/C2 levels from the Common European Framework of Reference for Languages[38] to ensure maximum comprehension of the English-spoken audio files used in all conditions. Participants were excluded if they reported having or having had a history of mental illnesses assessed via a pre-screening question, if they declared having meditated in the previous 6 months or if they did not match the English language proficiency required (participants had to be either native language level or fluent in English). Each participant was asked to take part in the survey using a smartphone with headphones or earphones attached, to ensure that participants could perform any of the mindfulness activities they were randomly assigned to (that is, mindful walking). Each site committed to collect between 70 and 120 participants; however, if a site collected fewer or more participants than was the target, we still used the data from those participants in the analysis. Each site collected a different number of participants, from a minimum of one and a maximum of 179. Our Rpubs page shows the total number of participants per site (https://rpubs.com/ale-sparacio92/920457). Data collection was performed blind to the experimental conditions but data analysis was not performed blind. However, given that all our analyses were preregistered, it is unlikely that the lack of blinding in data analysis introduced bias.

The dataset originally comprised 6,691 responses, including both the 'test answers' generated by the site collaborators while developing and previewing the survey and the actual answers submitted by the participants. From the initial participants in the survey, we excluded the following: 1,307 who self-identified as meditators or reported having engaged in meditation within 6 months before the experiment, 776 who did not meet the English language proficiency requirement and 981 who disclosed having a history of mental illnesses. Finally, 1,660 participants started the survey without using a smartphone with headphones attached. Among these participants who failed to meet the inclusion criteria, 1,491 simultaneously met several exclusion criteria. Respondents who did not meet one or more inclusion criteria ($n = 3,233$) were immediately directed towards the end of the survey

and we did not record further data from them. We also removed from analyses those who initiated the survey but did not progress up to the listening of the audio track ($n = 976$) and the 'test answers' provided by the collaborating researchers while developing the survey ($n = 19$); thus, the sample size dropped to $n = 2,463$. We then removed data from 19 participants who dropped out of the experiment and data from 205 participants who, according to our criteria, were considered careless respondents, yielding a final sample of 2,239 valid observations. Of these, 611 participants self-identified as male, 1,576 as female, 7 as transgender male, 2 as transgender female, 27 did not identify with any choice and 16 preferred not to say (mean age ($M_{age}$) = 22.4, s.d.$_{age}$ = 10.1; range 17–87; 94.2% students), with an approximately even distribution across the five experimental conditions ($n_{mindful walking}$ = 416, $n_{mindful breathing}$ = 469, $n_{loving kindness}$ = 427, $n_{body scan}$ = 449, $n_{book chapter control}$ = 478). We are not aware of how many participants were invited to the survey but declined to participate.

### Dealing with careless responders

We applied a set of rules to deal with responders[39] who were careless or had made insufficient effort, to reduce the random variance component in the data. First, we made the answers for the questions connected to our exclusion criteria (meditation experience, English language proficiency and mental illnesses) compulsory. For the questionnaires related to our dependent variables/moderator, we alerted respondents about unanswered questions but they had the possibility to continue with the survey without providing a response. Second, the programmed survey prevented participants from skipping the 15 min audio file (for both mindfulness exercises and control conditions) by blocking the screen with the audio of the meditation/control condition for 14 min, so as not to allow participants to proceed to the following survey page until the meditation was finished. Third, we identified and excluded participants who provided identical responses to a long series of items (that is, always selecting the answer 'strongly agree') by performing a long-string analysis. Using long-string analyses, we excluded participants with a string of consistent responses equal to or greater than 10 (that is, half of the scale length).

### Distribution of participants across sites

Thirty-seven sites participated in the data collection (see the full list at https://osf.io/uh3pk). Participants could be recruited through the SONA system (the platform used to recruit student participants from universities, https://www.sona-systems.com/) of the respective institution or via crowdsourcing platforms such as mTurk or Prolific. Participants could come from any geographic area if they met our inclusion criteria and could be given either credits or financial compensation in exchange for participating in the study.

### Materials

**Self-administered mindfulness interventions.** To compile a list of self-administered mindfulness exercises to be tested in our multi-site project, we initially conducted a survey among mindfulness practitioners, whom we asked to recommend the most prominent and widely used exercises in their practice. We then retained the most popular exercises suggested by the surveyed practitioners, which we cross-referenced with the exercises included by Matko[40] in an inventory of present popular mindfulness exercises. This combined approach led to the selection of four types of mindfulness exercises: body scan, mindful breathing, mindful walking and loving kindness meditation. The full procedure that led us to the selection of the four self-administered mindfulness exercises can be found in the extended preregistration document.

The four audio files of the mindfulness exercises and the three audio files of the stories of the non-mindful active control condition were recorded by the same certified meditation trainer, C. Spiessens, a BAMBA registered mindfulness teacher in MBSR (https://www.christophspiessens.com/) and each lasted 15 min. The exact text of the seven

meditations and of the three stories used in the active control condition can be found on our OSF project page (https://osf.io/6w2zm/). The seven recordings can be found on the Soundcloud page of the project (https://soundcloud.com/listening-385769822).

**Mindfulness conditions.** In body scan, the meditation trainer invited participants to 'scan' their parts of the body. Every time the mind wandered, the meditation trainer invited participants to bring back the awareness and attention to the part of their body they were 'scanning'. During mindful breathing, the meditation trainer invited participants to 'stay with their breath', without changing the way they were breathing. When their mind wandered, the meditation trainer invited participants to bring their attention back to their breath with kindness and patience. During the loving kindness meditation, the trainer encouraged participants to direct loving kindness toward themselves and then to extend these feelings of loving kindness towards somebody else. During mindful walking, the meditation trainer asked participants to walk in a quiet place (preferably indoors or in a place as isolated as possible from distractions), while listening to the instructions. During this practice, the meditation trainer invited participants to bring their awareness to the experience of walking and subsequently the meditation trainer invited them to 'feel' the physical sensations of contact of their feet with the ground.

**Control conditions.** Participants in the active control condition listened to an excerpt from 'Silverview' by John le Carré[21] (word count 1,838), 'The Old Man and the Sea' by Ernest Hemingway[41] (word count 2,039) or 'Smith of Wootton Major' by J. R. R. Tolkien[22] (word count 2,309). We used more than one story excerpt to increase the variance of the control conditions and thus push towards greater generalizability across stimuli[42]. These three excerpts had a similar word count, were written in standard English, did not feature major plot changes and were thus unlikely to elicit strong emotions. Participants had equal chances of listening to any one of the three story excerpts.

**Neuroticism.** We measured this trait with the neuroticism subscale of the International Personality Item Pool five NEO domains, comprising 20 items[43]. Examples of items include 'I often feel blue' or 'I am filled with doubts about things' and answers ranged from 1 (very inaccurate) to 5 (very accurate; coefficient omega $\omega_u = 0.90$).

**Stress.** Participants answered the 20 item STAI Form Y-1 (ref. [19]). They indicated how they felt in that exact moment on 20 items (for example, 'I am tense'; 'I feel frightened'; $\omega_u = 0.92$) on a 4-point scale (1, not at all; 2, somewhat; 3, moderately so; 4, very much so). By using the STAI Form Y-1 scale, we aimed to measure the short-term effects of stress on individuals. This scale, after all, has been shown to correlate with biomarkers of stress in previous research (salivary α-amylase[44]).

**Emotion dimensions.** Participants filled in the self-assessment manikin scale, a three-item non-verbal pictorial assessment technique which measures emotions on three different dimensions, namely pleasure, arousal and dominance[45]. The self-assessment manikin scale is the picture-oriented version of the widely used semantic differential scale[46]. This instrument measures the three-dimensional structure of stimuli, objects and situations with 18 bipolar adjective pairs which can be rated along a 9-point scale. This measure was not the primary dependent variable of our study but we added it in the study for the exploratory analyses.

**Demographics.** Participants provided information regarding their age, gender, country of birth, country of residence, whether they were students or not, which university they were studying at (for the former) and what was their current occupation (for the latter).

## Simulation of the sequential Bayesian design

Before the data collection, we simulated data based on a Bayes factor design analysis to assess the expected efficiency and informativeness of the present design. The aim of the simulation was to establish (1) the expected likelihood of the study to provide compelling relative evidence either in favour of $H_0$ (BF$_{10}$ = 1/10) or $H_1$ (BF$_{10}$ = 10), (2) the likelihood of obtaining convincing but misleading evidence and (3) the likelihood that the study points into the correct direction even if stopped earlier due pragmatic constraints on sample size[47].

Given these aims, we modelled a sequential design with a maximum $n$ where the data collection continues until either the threshold for compelling evidence is met or the maximum $n$ is reached. Although 41 laboratories indicated an interest in the project, we took the conservative estimate of 30 data-collecting laboratories. Each laboratory was expected to collect data of at least $n = 70$ participants, with a maximum $n$ at 120 (translating to minimum 420 and maximum 720 participants per condition). Our goal was to be able to detect an effect size of $d = 0.20$; we modelled the true value to vary between laboratories by repeatedly (for each simulation) drawing from a normal distribution, $\delta \sim n(0.20, 0.05)$, with a 95% probability that the effect size falls between $d = 0.10$ and 0.30.

We tested the effectiveness of four standalone interventions using a between-participants adaptive group design, whereupon hitting a threshold of compelling evidence in one condition, we planned to allocate the rest of the participants into other conditions where the threshold had not been met yet. The simulation, however, assumed a conservative scenario with equal $n$ across all conditions, therefore, simplifying the computations to a single between-participants $t$-test scenario.

The results (Fig. 3) show that, given the assumed design, the probability of the test arriving at the boundary of compelling evidence (BF$_{10}$ = 10 or 1/10) was 0.79 (0.72 at $H_1$ and 0.07 erroneously at $H_0$). The probability of terminating at a maximum $n$ of 720 per condition was 0.21; 0.05 of showing some evidence for $H_1$ (BF$_{10}$ > 3), 0.13 of being inconclusive (3 > BF$_{10}$ > 1/3) and 0.03 of showing evidence for $H_0$ (BF$_{10}$ < 1/3). For the test of a single condition against controls, the sequential design is expected to be 27% more effective than collecting a fixed maximum $n$ per laboratory, with the average $n$ at the stopping point (BF boundary and maximum $n$) at 526. Even conservatively assuming a balanced-$n$ situation, the informativeness of the design thus appeared to be adequate and the use of the adaptive design would probably enhance informativeness and/or resource efficiency.

## Procedure

Participants accessed the experiment via a Qualtrics link. We provided participants with detailed information about the study (see 'Participants information sheet' included in the IRB package, https://osf.io/6w2zm/) and asked for their consent to participate. We asked them to use a smartphone with headphones or earphones attached instead of a computer or laptop. We asked participants whether they started the survey from a device other than a smartphone; if they answered positively, we asked them to exit the survey and to restart it, this time using a smartphone with headphones or earphones attached. We then asked participants to sit in a quiet place such as a room where they would not be disturbed for 20 min. After providing informed consent, participants completed the neuroticism measure, then were randomly allocated by the Qualtrics algorithm to one of the four intervention conditions or one of the three control conditions, each lasting 15 min. On completion, participants answered the main study outcome, namely the stress measure and the self-assessment manikin scale. Finally, participants provided demographic information, were then thanked and debriefed and were awarded credit or payment depending on the site policy.

## Analysis plan

To assess the effectiveness of the chosen mindfulness exercises against the control conditions at reducing stress in participants in an efficient manner, we carried out four independent-samples Bayesian $t$-tests to

determine whether there was a difference between each mindfulness exercise and the active control condition. This study was originally conducted as a sequential Bayesian design[48]. The data were continuously monitored to see when each condition met the compelling evidence threshold of $BF_{10}$ of 10 in favour of $H_1$ or a $BF_{10}$ of 1/10 in favour of $H_0$. When we monitored the data, three out of four mindfulness exercises reached the $BF_{10}$ threshold of 1/10 in favour of $H_0$ before reaching the $BF_{10}$ of 10 in favour of $H_1$ as the sample increased. A detailed explanation of the sequential Bayesian design can be found in the extended preregistration document on the OSF page at https://osf.io/us5ae.

We used a two-tailed test using a non-informative Jeffreys–Zellner–Siow Cauchy prior for the alternative hypothesis with a default $r$-scale of $\sqrt{2}/2$ (ref. 49). To account for the hierarchical nature of the data, we compared the condition means using a Bayesian mixed-effects model which involved a random intercept for the site and for the different stories used in the non-mindful active control condition. We set our threshold of compelling evidence on the basis of which we would have drawn inferences about the results: a Bayes factor ($BF_{10}$) of 10 in favour of $H_1$ or a Bayes factor of 1/10 favoring $H_0$. We chose a Bayes factor of 10 because, according to the classification of ref. 20, it demarcates the threshold between moderate and strong evidence. Here, using a Bayes factor of 10, we aimed to substantially decrease the probability of misleading evidence[48]. In the Bayesian analyses, we only engaged in comparative inference using Bayes factors (comparing the likelihood of the data under two competing hypotheses, $H_0$ and $H_1$) and for this reason we did not estimate posteriors. Finally, we decided not to screen for and exclude outliers and we did not perform any (nonlinear) transformations contingent on the observed data.

### Exploratory analyses
We also carried out analyses exploring the effect of the experimental conditions on pleasure, arousal and dominance and for the moderating effect of neuroticism. We performed separate Bayesian $t$-tests for each dimension of the self-assessment manikin scale (pleasure, arousal and dominance) comparing our experimental conditions with the control condition. We then looked at the Bayes factor to establish whether the data favoured $H_1$ or $H_0$. We compared the means of the different conditions using a Bayesian mixed-effects model with a random intercept for laboratory and for the different stories used in the non-mindful active control condition to account for the hierarchical nature of the data.

To examine whether neuroticism moderated the effects of the four experimental conditions on stress, we compared the model with the interaction to the model with only the main effects (using the lmBF function) and we reported the corresponding $BF_{10}$. If the model with the interaction was preferred to the model with only the main effects of a $BF_{10}$ of 10 or more, we regarded it as solid evidence of the moderation of neuroticism on stress. We performed a similar analysis to investigate the potential moderation of English language proficiency on stress levels. The analyses for the current project were performed using RStudio v.2023.09.0 + 463.

### Not preregistered analyses
Several analyses conducted in the 'exploratory analyses' section were not explicitly outlined in the preregistration. These additional analyses included the computation of heterogeneity and Cohen's $d$ for each condition when compared to the active control conditions and moderation effects by considering English language proficiency. Additionally, robustness analyses were incorporated at the reviewer's request.

### Reporting summary
Further information on research design is available in the Nature Portfolio Reporting Summary linked to this article.

### Data availability
This project was preregistered on OSF on 22 March 2022, before the enrolment of the first participant (registration https://doi.org/10.17605/OSF.IO/UF4JZ). On editorial request, we retroactively registered our project as a clinical trial on ClinicalTrials.gov (https://clinicaltrials.gov/study/NCT06308744). Our data are available on the OSF (https://osf.io/6w2zm/) and via the GitHub repository (https://github.com/alessandro992/A-large-multi-site-test-of-self-administered-mindfulness). The data are available under the terms of the Creative Commons Attribution 4.0 International license (CC BY 4.0).

### Code availability
The full analysis code is publicly available at https://github.com/alessandro992/A-large-multi-site-test-of-self-administered-mindfulness and on our OSF page (https://osf.io/6w2zm/).

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

## Acknowledgements

The preparation of this work was partly funded by Swansea University Strategic Partnerships Research Scholarships from School of Psychology, Swansea University awarded to G.M.J.-B., PRIMUS/24/SSH/017 and NPO 'Systemic Risk Institute' (LX22NPO5101) grants awarded to I.R. and NeuroCog 'Project MIBODA' from Université Grenoble Alpes awarded to H.I. R.M.R. was supported by the Australian Research Council (grant no. DP180102384) and the John Templeton Foundation (grant no. 62631). We also thank the SCREEN/MSH-Alpes platform for providing access to Qualtrics. The funders had no role in study conceptualization, design, data collection and analysis, decision to publish or preparation of the manuscript.

## Author contributions

A.S., G.M.J.-B and H.I. conceptualized this work. A.S., I.R. and F.G. undertook data curation. A.S., I.R. and F.G. were responsible for the formal analysis. H.I., I.R., R.M.R. and G.M.J.-B. acquired funding. B.N.U., J.L., T.T., S.J.D., J.L.D., J.S., J.D.P., R.M.R., Z.F., A.L., C.M.-K., M.B.F., K.S., C.C.W., W.C.H., B.M.S., S.K.S., G.R., P.T.F., V.J., A.K., P.Z., C.L.J., Y.K., M.A., C.F., M.F.B., H.B., A.B.E.B., M.D., C.E.N., J.C.B., C.M.G., J.M.B., S.M.G., S.D., W.E.D., T.J.W., W.B.M., J.L.H., L.S.R., M.G.S., S.S.D., S.P., S.S.-S., Z.I.W., M.V.D., S.B., A.H.L., C.E.H., L.V.O'B., T.U., J.S.M., K.L.v.d.S., H.B. and C.N.T.-W. undertook investigations. A.S., G.M.J.-B., H.I. and I.R. developed the methodology. A.S., G.M.J.-B. and H.I. were responsible for project administration. A.S., G.M.J.-B., H.I. and C.S. obtained resources. A.S. developed the software. A.S., G.M.J.-B., H.I. and I.R. supervised the project. A.S., G.M.J.-B., H.I. and I.R. undertook validation. A.S. produced the visualization. A.S., G.M.J.-B., H.I. and I.R. wrote the original draft. A.S., G.M.J.-B., H.I., I.R., R.M.R., S.J.D., G.R., W.B.M. and W.C.H. reviewed and edited the final paper. The certified mindfulness

instructor (C.S.) only recorded the meditations and was not involved in writing any parts of the paper, including the other stages of the project (for example, choice of the analyses, data analysis).

## Competing interests

H.I. is the director of a startup, Annecy Behavioral Science Lab, which seeks to promote human flourishing and social connection. He began in the startup when the manuscript was nearly finalized. The other authors declare no competing interests.

## Additional information

**Correspondence and requests for materials** should be addressed to Alessandro Sparacio, Hans IJzerman, Ivan Ropovik or Gabriela M. Jiga-Boy.

Alessandro Sparacio [1,2,3] ✉, Hans IJzerman [1,4,5] ✉, Ivan Ropovik [6,7,8] ✉, Filippo Giorgini [9], Christoph Spiessens[10], Bert N. Uchino[11], Joshua Landvatter[11], Tracey Tacana [11], Sandra J. Diller [12,13], Jaye L. Derrick [14], Joahana Segundo[14], Jace D. Pierce[14], Robert M. Ross [15], Zoë Francis [16], Amanda LaBoucane[16], Christine Ma-Kellams[17], Maire B. Ford[18], Kathleen Schmidt [19], Celia C. Wong[20], Wendy C. Higgins [15], Bryant M. Stone[21], Samantha K. Stanley [22], Gianni Ribeiro [23], Paul T. Fuglestad [24], Valerie Jaklin[25], Andrea Kübler[25], Philipp Ziebell [25], Crystal L. Jewell [26], Yulia Kovas [27], Mahnoosh Allahghadri[27], Charlotte Fransham[27], Michael F. Baranski[28], Hannah Burgess[28], Annika B. E. Benz[29], Maysa DeSousa[30], Catherine E. Nylin [31], Janae C. Brooks[31], Caitlyn M. Goldsmith[31], Jessica M. Benson [32], Siobhán M. Griffin [33], Stephen Dunne [34], William E. Davis [35], Tam J. Watermeyer[34,36], William B. Meese[37], Jennifer L. Howell [37], Laurel Standiford Reyes [38], Megan G. Strickland[39], Sally S. Dickerson[39], Samantha Pescatore[39], Shayna Skakoon-Sparling[40], Zachary I. Wunder[41], Martin V. Day [42], Shawna Brenton [42], Audrey H. Linden [43,44], Christopher E. Hawk [45], Léan V. O'Brien [46], Tenzin Urgyen[47], Jennifer S. McDonald[48], Kim Lien van der Schans[49], Heidi Blocker[50], Caroline Ng Tseung-Wong[46] & Gabriela M. Jiga-Boy [2] ✉

¹LIP/PC2s, Université Grenoble Alpes, Grenoble, France. ²School of Psychology, Swansea University, Swansea, UK. ³Singapore Institute for Clinical Sciences (SICS), A*STAR, Singapore, Singapore. ⁴Annecy Behavioral Science Lab, Saint-Jorioz, France. ⁵Institut Universitaire de France, Paris, France. ⁶Institute of Psychology, Czech Academy of Sciences, Prague, Czech Republic. ⁷Faculty of Education, Institute for Research and Development of Education, Charles University, Prague, Czech Republic. ⁸Centre of Social and Psychological Sciences, Slovak Academy of Sciences, Štefánikova, Bratislava, Slovakia. ⁹Department of Economics, Management and Statistics (DEMS), University of Milano-Bicocca, Milan, Italy. ¹⁰Spiessens Coaching Solutions Ltd, Manchester, UK. ¹¹Department of Psychology, College of Social and Behavioral Science, University of Utah, Salt Lake City, UT, USA. ¹²Private University Seeburg Castle, Seekirchen am Wallersee, Austria. ¹³LMU Munich, Munich, Germany. ¹⁴University of Houston, Houston, TX, USA. ¹⁵School of Psychology, Macquarie University, Sydney, New South Wales, Australia. ¹⁶University of the Fraser Valley, Abbotsford, British Colombia, Canada. ¹⁷San Jose State University, San José, CA, USA. ¹⁸Psychology department, Loyola Marymount University, Los Angeles, CA, USA. ¹⁹Ashland University, Ashland, OH, USA. ²⁰SUNY Brockport, Brockport, NY, USA. ²¹Department of Mental Health, Bloomberg School of Public Health, Johns Hopkins University, Baltimore, MD, USA. ²²Australian National University, Canberra, Australian Capital Territory, Australia. ²³School of Law and Justice, The University of Southern Queensland, Ipswich, Queensland, Australia. ²⁴University of North Florida, Jacksonville, FL, USA. ²⁵Department of Psychology, University of Würzburg, Würzburg, Germany. ²⁶Iowa State University, Ames, IA, USA. ²⁷Goldsmiths University of London, London, UK. ²⁸Pennsylvania Western University California, California, PA, USA. ²⁹University of Konstanz, Konstanz, Germany. ³⁰Springfield College, Springfield, MA, USA. ³¹Department of Social and Behavioral Sciences, Glendale Community College, Glendale, AZ, USA. ³²St. Olaf College, Northfield, MN, USA. ³³University of Limerick, Limerick, Ireland. ³⁴Northumbria University, Newcastle upon Tyne, UK. ³⁵Wittenberg University, Springfield, OH, USA. ³⁶University of Edinburgh, Edinburgh, UK. ³⁷University of California, Merced, Merced, CA, USA. ³⁸University of Southern Indiana, Evansville, IN, USA. ³⁹Pace University, New York, NY, USA. ⁴⁰Toronto Metropolitan University (formerly Ryerson), Toronto, Ontario, Canada. ⁴¹Wayne State University, Detroit, MI, USA. ⁴²Memorial University of Newfoundland, St. John's, Newfoundland, Canada. ⁴³Centre for Research in Autism and Education, Institute of Education, University College London, London, UK. ⁴⁴Department of Psychology and Counselling, The Open University, Milton Keynes, UK. ⁴⁵DigiPen Institute of Technology, Redmond, WA, USA. ⁴⁶University of Canberra, Canberra, Australian Capital Territory, Australia. ⁴⁷University of Northern Iowa, Cedar Falls, IA, USA. ⁴⁸Idaho State University, Pocatello, ID, USA. ⁴⁹Behavioural Science Institute, Radboud University, Nijmegen, the Netherlands. ⁵⁰Eastern Oregon University, La Grande, OR, USA. ✉e-mail: Alessandro_Sparacio@sics.a-star.edu.sg; hans@absl.io; ivan.ropovik@gmail.com; G.Jiga@swansea.ac.uk

# Reporting Summary

## Statistics

For all statistical analyses, confirm that the following items are present in the figure legend, table legend, main text, or Methods section.

| n/a | Confirmed | |
|---|---|---|
| ☐ | ☒ | The exact sample size (*n*) for each experimental group/condition, given as a discrete number and unit of measurement |
| ☐ | ☒ | A statement on whether measurements were taken from distinct samples or whether the same sample was measured repeatedly |
| ☐ | ☒ | The statistical test(s) used AND whether they are one- or two-sided<br>*Only common tests should be described solely by name; describe more complex techniques in the Methods section.* |
| ☐ | ☒ | A description of all covariates tested |
| ☐ | ☒ | A description of any assumptions or corrections, such as tests of normality and adjustment for multiple comparisons |
| ☐ | ☒ | A full description of the statistical parameters including central tendency (e.g. means) or other basic estimates (e.g. regression coefficient) AND variation (e.g. standard deviation) or associated estimates of uncertainty (e.g. confidence intervals) |
| ☐ | ☒ | For null hypothesis testing, the test statistic (e.g. *F*, *t*, *r*) with confidence intervals, effect sizes, degrees of freedom and *P* value noted<br>*Give P values as exact values whenever suitable.* |
| ☐ | ☒ | For Bayesian analysis, information on the choice of priors and Markov chain Monte Carlo settings |
| ☐ | ☒ | For hierarchical and complex designs, identification of the appropriate level for tests and full reporting of outcomes |
| ☒ | ☐ | Estimates of effect sizes (e.g. Cohen's *d*, Pearson's *r*), indicating how they were calculated |

*Our web collection on statistics for biologists contains articles on many of the points above.*

## Software and code

Policy information about availability of computer code

| Data collection | We used a Qualtrics survey to collect the data of our multi-site study. We did not use any software for the Data collection. |
|---|---|
| Data analysis | We used Rstudio version 2023.09.0+463 for the data analysis of the current project. |

For manuscripts utilizing custom algorithms or software that are central to the research but not yet described in published literature, software must be made available to editors and reviewers. We strongly encourage code deposition in a community repository (e.g. GitHub). See the Nature Portfolio guidelines for submitting code & software for further information.

## Data

Policy information about availability of data

All manuscripts must include a data availability statement. This statement should provide the following information, where applicable:
- Accession codes, unique identifiers, or web links for publicly available datasets
- A description of any restrictions on data availability
- For clinical datasets or third party data, please ensure that the statement adheres to our policy

Full data are publicly available at https://osf.io/6w2zm/ and https://github.com/alessandro992/A-large-multi-site-test-of-self-administered-mindfulness/blob/main/finaldata.csv

# Research involving human participants, their data, or biological material

Policy information about studies with <u>human participants or human data</u>. See also policy information about <u>sex, gender (identity/presentation), and sexual orientation</u> and <u>race, ethnicity and racism</u>.

| | |
|---|---|
| Reporting on sex and gender | We determined the gender of participants based on self-reporting methods. Participants could answer to which gender they identified the most, being given six different options (i.e., male, female, transgender male, transgender female, prefer not to say, and an open answer in which they could write their gender). We did not collect disaggregated sex and gender data.<br>We did not conduct sex- and gender-based analyses because the literature we had reviewed did not provide us with evidence to predict gender differences regarding mindfulness and stress reduction. |
| Reporting on race, ethnicity, or other socially relevant groupings | Participants reported their country of origin and the country where they currently lived. This information was only used to describe the sample; we have not conducted any analysis involving such information. |
| Population characteristics | See above. |
| Recruitment | The sites involved in this project recruited participants with a Qualtrics link that was provided to them by the main investigator. The sites' coordinators were told by the main investigator that participants could be recruited using the SONA system of their respective institution or via crowdsourcing platforms such as mTurk or Prolific Academic. While using a combination of unpaid (e.g., SONA) and paid (e.g., Prolific) participation platforms could have mitigated self-selection bias, conducting the experiment solely online still limited our ability to completely eliminate this bias. Just like with practically any randomized controlled trial (RCT) on human subjects, self-selection of participants is inevitable. Participants, patients, etc., can decide whether they want to take part in the study. They can do so before and at any time during the experiment. Self-selection is critical when the goal is to describe a population (e.g., prevalence studies). It is, however, not a threat to the integrity of the results when the goal is to establish causal knowledge because, by definition, causal inference in RCTs is comparative, where we want to examine evidence for relative treatment effectiveness (Msaouel et al., 2023). The goal of an RCT is thus not to arrive at particular statements about the current state of the population, but rather identify and disentangle causal mechanisms. These principles are then likely transportable to the members of the given population, and frequently even beyond (Bradburn et al., 2020). |
| Ethics oversight | The study first received ethical approval from Swansea University's School of Psychology Research Ethics Sub-committee, while the sites that participated in the data collection either received ethical approval from their local IRBs or stated that they were exempt (e.g., if their IRB accepted the ethics approval awarded by Swansea University and did not request the local collaborator to submit their own application). Each site's IRB protocols with ethics details and acceptance of each protocol can be found on the OSF project page at https://osf.io/6w2zm/. Swansea University and Université Grenoble Alpes carried the administrative organization for the study. Swansea University was also the data controller for this project. The personal data of participants were processed for the purposes outlined in the information sheet (see the document Information Sheet at https://osf.io/xuznc/). Standard ethical procedures involved participants providing their consent to participate in this study by completing the consent form that was administered at the beginning of the online survey used for the experiment. |

Note that full information on the approval of the study protocol must also be provided in the manuscript.

# Field-specific reporting

Please select the one below that is the best fit for your research. If you are not sure, read the appropriate sections before making your selection.

☐ Life sciences    ☒ Behavioural & social sciences    ☐ Ecological, evolutionary & environmental sciences

For a reference copy of the document with all sections, see <u>nature.com/documents/nr-reporting-summary-flat.pdf</u>

# Behavioural & social sciences study design

All studies must disclose on these points even when the disclosure is negative.

| | |
|---|---|
| Study description | The current multi-site project followed the route of a parallel randomized controlled trial. |
| Research sample | The study included participants from Australia, Europe, the UK, Canada, and the US. We had three exclusion criteria:<br>1) Participants had to be current non-meditators or to have not meditated in the 6 months prior to the experiment,<br>2) Participants had to be fluent or native English speakers, and<br>3) Participants had to declare they had not had a history of mental illness.<br>Criterion 1 was used because the experiment focused on the effects of single brief exercises on non-meditators to better understand the potential benefits of mindfulness practices for this population. Criterion 2 was used because the audio files used were recorded in English. Criterion 3 was used because previous research has shown that mindfulness interventions have at times resulted in psychotic episodes, panic attacks, and depersonalization; thus, we needed to screen out participants for whom the mindfulness intervention could have been detrimental. |

The final sample was not representative. After excluding participants that did not fit our inclusion criteria, we retained 2,239 valid observations (of these, 611 self-identified as males, 1,576 as females, seven as transgender males, two as transgender females, 27 did not identify with any choice, 16 preferred not to say; Mage = 22.4, SDage = 10.1; range 17-87; 94.2% students).

The rationale behind selecting the study sample (i.e., participants who were non-meditators, fluent in English and had no history of mental illness) was to specifically examine the effects of brief self-administered mindfulness interventions in isolation from potential confounds. A previous experience of engaging with meditation could have changed the baseline for any potential mindfulness effects; non-fluent English language proficiency could have created problems with mindfulness instruction comprehension; and a history of mental illnesses might have exacerbated potential meditation-related adverse effects (Britton, Lindahl, Cooper, Canby & Palitsky, 2021). This study aimed to deepen our understanding of how mindfulness practices can benefit individuals who are new to mindfulness. Although the sample is not representative of the general population, it provides valuable insights into the unique impacts of mindfulness on stress reduction measured experimentally, and practical relevance of mindfulness techniques among a predominantly young, student demographic spread across various geographic regions.

| | |
|---|---|
| Sampling strategy | Each site collected the data using Qualtrics that redirected participants to the same survey; however, the URL address was tailored for each data-collecting site to allow recording the site participants belonged to. Participants were randomly allocated to the experimental conditions or control using Qualtrics' random block function. Prior to data collection, we conducted a simulation based on a Bayes Factor Design Analysis (BFDA) to assess the expected efficiency and informativeness of our study design. The simulation aimed to determine (1) the expected likelihood of the study to provide compelling evidence either in favor of H0 (BF01 = 1/10) or H1 (BF10 = 10), (2) the likelihood of obtaining convincing but misleading evidence, and (3) the likelihood that the study points in the correct direction even if stopped earlier due to pragmatic constraints on sample size. We modeled a sequential design with a maximum N of 720 and a minimum sample size of 420 participants per condition, with a goal of detecting an effect size of d = 0.20. We tested four interventions using a between-participants adaptive group design and found that the probability of arriving at compelling evidence was .79. A more detailed explanation of this simulation can be found at p. 10 of the manuscript. |
| Data collection | The experiment was conducted entirely online, and participants were instructed to complete it in a quiet environment for 20 minutes. Participants were asked to access the experiment using a desktop or laptop computer and not a mobile smartphone because one condition of the experimental design involved mindful walking, so we needed to ensure that any participant would be able to complete that task, if they were (randomly) distributed in that condition. The researchers were blind to the experimental conditions the participants were allocated to because the allocation was done using Qualtrics' randomizer function. |
| Timing | The data collection started on March 23rd, 2022 and finished on June 30th, 2022. |
| Data exclusions | The dataset originally comprised 6,691 responses, including both the 'test answers' generated by the site collaborators while developing and previewing the survey, and the actual answers submitted by the participants. From the survey's initial participants, we excluded the following: 1,307 who self-identified as meditators or reported having engaged in meditation within six months prior to the experiment, 776 who did not meet the English language proficiency requirement, and 981 who disclosed having a history of mental illnesses. Finally, 1660 participants started the survey without using a smartphone with headphones attached. Among these participants who failed to meet the inclusion criteria, 1,491 simultaneously met multiple exclusion criteria. Respondents who did not meet one or more inclusion criteria (N = 3, 233) were immediately directed towards the end of the survey, and we did not record further data from them. We also removed from analyses those who initiated the survey but did not progress up to the listening of the audio track (N = 976), and the 'test answers' provided by the collaborating researchers while developing the survey (N = 19); thus, the sample size dropped to N = 2,463. We then removed data from 19 participants that dropped out of the experiment and data from 205 participants who, according to our criteria, were considered careless respondents, yielding a final sample of 2,239 valid observations. Of these, 611 participants self-identified as male, 1,576 as female, seven as transgender male, two as transgender female, 27 did not identify with any choice, and 16 preferred not to say (Mage = 22.4, SDage = 10.1; range 17-87; 94.2% students), with an approximately even distribution across the five experimental conditions (Nmindful walking = 416, Nmindful breathing = 469, Nloving - kindness = 427, Nbody scan = 449, Nbook chapter – control = 478). We are not aware of how many participants were invited to the survey, but declined to participate. |
| Non-participation | We did not collect data from participants who declined to provide consent, as they were led to the end of the survey. However, participants who began the experiment but dropped out before the set of responses related to the main dependent variable were categorized as "careless participants" and were excluded from the main analyses. We do not have information regarding the reasons why participants may have abandoned the experiment. |
| Randomization | Participants were randomized to one of the experimental conditions (1,2,3,4) or to one of the active control conditions (story a,b,c). As an example involving 15 participants, this is how they were expected to be randomized by the Qualtrics software: <br> Condition 1 Body-scan : 3 participants <br> Condition 2 Loving kindness: 3 participants <br> Condition 3 Mindful breathing: 3 participants <br> Condition 4 Mindful walking: 3 participants <br> Condition 5 control condition story a: 1 participant <br> Condition 5 control condition story b: 1 participant <br> Condition 5 control condition story c: 1 participant |

# Reporting for specific materials, systems and methods

We require information from authors about some types of materials, experimental systems and methods used in many studies. Here, indicate whether each material, system or method listed is relevant to your study. If you are not sure if a list item applies to your research, read the appropriate section before selecting a response.

## Materials & experimental systems

| n/a | Involved in the study |
|---|---|
| ☒ | ☐ Antibodies |
| ☒ | ☐ Eukaryotic cell lines |
| ☒ | ☐ Palaeontology and archaeology |
| ☒ | ☐ Animals and other organisms |
| ☐ | ☒ Clinical data |
| ☒ | ☐ Dual use research of concern |
| ☒ | ☐ Plants |

## Methods

| n/a | Involved in the study |
|---|---|
| ☒ | ☐ ChIP-seq |
| ☒ | ☐ Flow cytometry |
| ☒ | ☐ MRI-based neuroimaging |

## Clinical data

Policy information about clinical studies

All manuscripts should comply with the ICMJE guidelines for publication of clinical research and a completed CONSORT checklist must be included with all submissions.

| | |
|---|---|
| Clinical trial registration | https://clinicaltrials.gov/study/NCT06308744 |
| Study protocol | The full protocol of the study can be found at https://osf.io/uf4jz (Registration DOI: https://doi.org/10.17605/OSF.IO/UF4JZ) and in our ClinicalTrials.gov page (https://clinicaltrials.gov/study/NCT06308744). |
| Data collection | Data collection for the study occurred from March 23rd to June 30th, 2022. This study was conducted in a fully decentralized manner, meaning that participants did not visit a laboratory for data collection. Instead, participants engaged with the experiment remotely (e.g., from their home) through a provided Qualtrics link. |
| Outcomes | For our primary outcome measure, we utilized the 20-item State-Trait Anxiety Inventory, Form Y-1 (STAI19) to evaluate the immediate stress responses of the participants. They were asked to express their current feelings through 20 specific statements, such as "I am tense" and "I feel frightened," employing a 4-point scale ranging from "Not at all" to "Very much so." This scale's utility is underpinned by its established correlation with stress biomarkers, like salivary $\alpha$-amylase, in prior studies, thereby providing a robust measure of the short-term effects of stress. As a secondary outcome measure, we incorporated the Self-Assessment Manikin scale, a non-verbal, pictorial tool that assesses emotional responses across three dimensions: pleasure, arousal, and dominance. Though not the primary focus of our research, this scale was included for exploratory analysis to enrich our understanding of the participants' emotional states after the listening of the audio track. |

## Plants

| | |
|---|---|
| Seed stocks | *Report on the source of all seed stocks or other plant material used. If applicable, state the seed stock centre and catalogue number. If plant specimens were collected from the field, describe the collection location, date and sampling procedures.* |
| Novel plant genotypes | *Describe the methods by which all novel plant genotypes were produced. This includes those generated by transgenic approaches, gene editing, chemical/radiation-based mutagenesis and hybridization. For transgenic lines, describe the transformation method, the number of independent lines analyzed and the generation upon which experiments were performed. For gene-edited lines, describe the editor used, the endogenous sequence targeted for editing, the targeting guide RNA sequence (if applicable) and how the editor was applied.* |
| Authentication | *Describe any authentication procedures for each seed stock used or novel genotype generated. Describe any experiments used to assess the effect of a mutation and, where applicable, how potential secondary effects (e.g. second site T-DNA insertions, mosiacism, off-target gene editing) were examined.* |

