## [Peer Review File · Nature Human Behaviour]

Peer Review Information

Journal: Nature Human Behaviour

Manuscript Title: Self-administered mindfulness interventions reduce stress in a large, randomized controlled multi-site study

Corresponding author name(s): Alessandro Sparacio, Hans IJzerman, Ivan Ropovik, Gabriela Jiga-Boy

Editorial Notes:

Parts of this Peer Review File have been redacted as indicated to remove third-party material?

Reviewer Comments & Decisions:

Decision Letter, initial version:

28th June 2023

Dear Dr Sparacio,

Thank you once again for your manuscript, entitled "A large, multi-site test of self-administered mindfulness, effects on stress regulation among English speakers", and for your patience during the peer review process.

Your Article has now been evaluated by 3 referees. You will see from their comments copied below that, although they find your work of potential interest, they have raised quite substantial concerns. In light of these comments, we cannot accept the manuscript for publication, but would be interested in considering a revised version if you are willing and able to fully address reviewer and editorial concerns.

We hope you will find the referees' comments useful as you decide how to proceed. If you wish to submit a substantially revised manuscript, please bear in mind that we will be reluctant to approach the referees again in the absence of major revisions. We are committed to providing a fair and constructive peer-review process. Do not hesitate to contact us if there are specific requests from the reviewers that you believe are technically impossible or unlikely to yield a meaningful outcome.

To guide the scope of the revisions, the editors discuss the referee reports in detail within the team, including with the chief editor, with a view to (1) identifying key priorities that should be addressed in revision and (2) overruling referee requests that are deemed beyond the scope of the current study. We hope that you will find the prioritised set of referee points to be useful when revising your study. Please do not hesitate to get in touch if you would like to discuss these issues further.

Specifically, we ask you to address Reviewer 1's concerns about the robustness of the methodology and potential confounds. These are fundamental and important concerns, which we shared also with the other reviewers. While more positive, Reviewer 3 also recognizes the potential weaknesses that Reviewer 1 mentions. In the light of their feedback, we ask that in your revised manuscript, you provide all methodological details, as well as ideally conduct additional robustness checks, and explain why your findings are not confounded and meaningful.

If you wish to submit a suitably revised manuscript, we would hope to receive it within 4 months. I would be grateful if you could contact us as soon as possible if you foresee difficulties with meeting this target resubmission date.

- Include a "Response to the editors and reviewers" document detailing, point-by-point, how you addressed each editor and referee comment. If no action was taken to address a point, you must provide a compelling argument. When formatting this document, please respond to each reviewer comment individually, including the full text of the reviewer comment verbatim followed by your response to the individual point. This response will be used by the editors to evaluate your revision and sent back to the reviewers along with the revised manuscript.
- Highlight all changes made to your manuscript or provide us with a version that tracks changes.

[REDACTED]

Thank you for the opportunity to review your work. Please do not hesitate to contact me if you have any questions or would like to discuss the required revisions further.

Sincerely,

[REDACTED]

Reviewer expertise:

Reviewer #1: mindfulness interventions

Reviewer #2: Bayesian analyses; mindfulness research

Reviewer #3: large-scale / multi-site studies and RCTs; mindfulness

REVIEWER COMMENTS:

Reviewer #1:

Remarks to the Author:

The manuscript describes a large multi-site study on possible effects of a one-time brief mindfulness practice. Unfortunately, there are too major limitations of this study.

Overall, the manuscript is not well structured. E.g., in the introduction, mindfulness and meditation are only defined or explained at p. 4 after several paragraphs on the popularity of apps. Another example is more serious: no rationale is provided in the introduction for examining moderator effect of neuroticism. It is provided in the Results section, together with the measuring instrument that is used, without a reference. In the Methods section, the instrument is not described well either, while the reference is missing altogether.

However, even more importantly: the mindfulness practices used are not comparable to the control conditions. It is clear that control conditions were not matched to meditation conditions on important characteristics such as word count, type of voice, emotionality of content etc. Listening to the conditions using the link provided by the authors it becomes clear that the control conditions (a) have much more text and no periods of silence in between, (b) are spoken by a different person than the meditation conditions, (c) using a louder volume, and (d) with references to tension/anxiety, which may have induced such emotions in listeners (e.g. in the excerpt of Silverview, the main character is experiencing anxiety). Therefore, any differences found may easily be due to one or more of these characteristics, rendering the results completely useless.

In addition, it is unclear how participants were recruited; In the information letter, the aim of the study is disclosed (study on how participants would feel after meditation), which likely introduced bias. As a consequence, it was pointless for me reading the discussion.

Minor points

Abstract misses information on participants (at least age and sex), standard deviations of the reported means. Also $d = .56$ is not small, but medium.

Table 1 is not clear: what do the M (SD) represent ("comparison" is not specific enough), a Note is needed to explain BF10

Unclear abbreviations (e.g., SONA)

The section on simulation of Bayesian is difficult to follow and seems unnecessary altogether (including Figure 1), except a brief statement.

The Bayesian analysis is difficult to understand or follow and given the large sample size and simple model, a more conventional analysis would be preferred to most readers.

Reviewer #2:

Remarks to the Author:

The manuscript presents a comprehensive multi-site study on self-administered mindfulness and its effects on stress regulation among English speakers. The study is well-structured, provides valuable insights, and showcases an excellent use of Bayesian statistics. There are only a few areas that could benefit from further clarification and discussion.

Exclusion of Participants: Participants were excluded if they reported currently having or having had a history of mental illnesses. However, it might be important to specify what kind of mental illnesses were considered for exclusion.

Moderation Analyses: The exploratory analyses investigated whether neuroticism and English level moderated the effects of mindfulness exercises on stress. However, the results of these analyses were inconclusive. It might be worth discussing potential reasons for these inconclusive results and how future research could further investigate these potential moderating effects.

I appreciate the authors' focus on brief mindfulness meditation in this study. In my own research, I have often encountered skepticism about the efficacy of brief meditation sessions in influencing cognitive and social cognitive outcomes. This study provides valuable evidence that even short periods of mindfulness practice can have a significant (but temporary) impact. I would encourage the authors to further emphasize this point in their discussion. The potency of brief meditation sessions, as demonstrated in this study, could be a powerful counter-argument to the notion that longer sessions are always necessary. Highlighting this could not only strengthen the current study but also contribute to a broader understanding in the field about the potential of brief mindfulness interventions.

I would like to draw the authors' attention to Figure 3 in the manuscript. While the figure provides important information, its current visual quality could be improved for better readability. Specifically, some of the numbers and site names are difficult to read due to overlapping elements. This overlap obscures important information and could potentially confuse readers. Additionally, the legend in the lower right corner could benefit from clearer and more consistent naming conventions. Lastly, the overall sharpness of the figure appears to be low, which further hampers readability. I would recommend revising this figure to enhance its clarity and effectiveness in conveying the intended information.

Reviewer #3:

Remarks to the Author:

This study examines four different self-administered mindfulness interventions comparing them to active control condition. The study is rather well-conducted and the subject is very interesting. There are however some things that could be improved.

It is stated that "Participants were excluded if they reported currently having or having had a history of mental illnesses assessed via a pre-screening question.." How was this asked from the participants and what is included in mental illnesses here?

Where and how did the sites collect the participants? Through which channels?

In the measures, please mention the number of items of each measure. For instance, the number of neuroticism items is missing.

On page 13, it is reported that the participants filled in the self-assessment Manikin scale. This scale is again referred to on page 14 under the heading Exploratory analysis. However, the analyses on this scale are not presented in the results nor discussed in the discussion section. It is unclear why this measure is mentioned at all.

On page 13, it is reported that "We asked students to indicate in which university they were studying and asked non-students their current occupation" Does this mean that there were no other students but university students? Why?

The moderation effects of neuroticism are examined by merging the different mindfulness conditions. However, there could be personality related differences in what kind of mindfulness training is most effective. It would be interesting to see the results for individual mindfulness tasks as well.

Please, remove meta-text from the discussion. This is such a short text that meta-text is not needed. (E.g. page 21 starting from line 455, page 22 starting from line 481) I would also recommend removing the heading "Strengths of the current study" since this seems to be a general discussion of the results, which is missing elsewhere.

I would recommend adding more discussion on the relevance and meaning of the results. Why is this study important? What is the practical meaning of these results? Are there recommendations that the authors give based on their results and previous literature? What is the most important take home message of this study?

While discussing generalizability the authors use the word generality. Generalizability would seem more suitable a term.

The participants were not randomly selected. Thus, there may be bias, that limits the generalizability of the findings. This should be discussed.

The authors mention (page 23, starting on line 518) that "These interventions should be intended as being effective in the short-term and are unlikely to affect dispositional traits (such as chronic stress). " It is no doubt true that a short intervention is unlikely to induce permanent changes at least if the mindfulness practice is not continued after the intervention. However, I would like to see the authors commenting on conditions in which mindfulness exercises could induce longer-term effects and what would be optimal timing for mindfulness practices. It has been suggested that long-term mindfulness practice that is started before the stress occurs, may be more effective in regulating stress, as compared to mindfulness practice that is started after the acute stress situation is already on. If one is new to mindfulness practice, it may be difficult to learn and concentrate on it, while simultaneously trying to cope with acute stress. A more effective approach could be to start practicing while the stress is not on and be prepared beforehand before the stress is encountered. This kind of discussion on optimal way of doing mindfulness practice would be important and give needed depth to the

discussion. This would give a more comprehensive picture about the effectiveness of mindfulness in stress reduction.

Author Rebuttal to Initial comments

Definition and Structure of the Manuscript:

- **Reviewer 1, comment 1:** The manuscript describes a large multi-site study on possible effects of a one-time brief mindfulness practice. Unfortunately, there are too major limitations of this study. Overall, the manuscript is not well structured. E.g., in the introduction, mindfulness and meditation are only defined or explained at p. 4 after several paragraphs on the popularity of apps.
- **Reviewer 2, comment 1:** The manuscript presents a comprehensive multi-site study on self-administered mindfulness and its effects on stress regulation among English speakers. The study is well-structured, provides valuable insights, and showcases an excellent use of Bayesian statistics. There are only a few areas that could benefit from further clarification and discussion.
- **Reviewer 3, comment 1:** This study examines four different self-administered mindfulness interventions comparing them to active control condition. The study is rather well-conducted and the subject is very interesting. There are however some things that could be improved.

Authors' Response: We thank the reviewers for their constructive feedback. To provide a conciliatory response regarding the manuscript organization that builds up on the praise it received but addresses the concerns raised, we carefully considered the initial manuscript's structure and: 1) incorporated a definition of mindfulness at the outset of both the abstract (Page 2) and the introduction (Page 3), 2) streamlined the manuscript's introduction to provide a more substantial argument for the need for testing the effects of individual mindfulness exercises (Page 3), and 3) added a more detailed rationale for using neuroticism as potential moderator (Page 4). We believe that the current manuscript introduction offers an all-round clearer rationale for the multi-site study design.

Rationale and Methods:

- **Reviewer 1 Comment 2:** Another example is more serious: no rationale is provided in the introduction for examining moderator effect of neuroticism. It is provided in the Results section, together with the measuring instrument that is used, without a reference.

In the Methods section, the instrument is not described well either, while the reference is missing altogether.

- **Reviewer 3 Comment 2:** In the measures, please mention the number of items of each measure. For instance, the number of neuroticism items is missing.

Authors' Response: We thank the reviewers for their feedback. The section detailing the moderation of neuroticism had previously appeared on Page 17 only. In the current manuscript, the rationale for this moderator is detailed on Page 4 (second paragraph). The revised text now reads as follow:

Finally, the potential moderating influence of different personality traits on the effects of these exercises remains largely unexplored. Previous research has indicated that neuroticism may moderate the psychological effects of mindfulness training^{14,15}. A meta-analysis appraising the evidence of 29 studies found that neuroticism exhibits the most pronounced association with self-reported individual differences in mindfulness among the Big Five personality traits ($r = -.45$ ¹⁶). Furthermore, one study found that individuals who scored higher in neuroticism displayed a more significant decrease in psychological distress and improvement of overall well-being when compared to a control group after participating in a Mindfulness-Based Stress Reduction (MBSR). While this study suggested that neuroticism moderated the effect, the power of the design (with $N = 244$) to detect smaller, but still theoretically meaningful interaction was modest¹⁷ and the authors acknowledged that the use of four possible moderators for each outcome may have inflated Type 1 errors¹⁴.

While the original manuscript had already included the neuroticism scale with example items, information regarding the scale reliability and its respective reference (Page 11), we have now added the number of items included in this scale (Page 10), resulting in the following description:

“We measured this trait with the Neuroticism subscale of the IPIP - 5 NEO domains, comprising 20 items⁴³. Examples of items include “I often feel blue” or “I am filled with doubts about things” and answers ranged from 1 (*Very inaccurate*) to 5 (*Very accurate*; $\omega_u = 0.90$)”.

- **Reviewer 1 Comment 3:** However, even more importantly: the mindfulness practices used are not comparable to the control conditions. It is clear that control conditions were not matched to meditation conditions on important characteristics such as word count, type of voice, emotionality of content etc. Listening to the conditions using the link provided by the authors it becomes clear that the control conditions (a) have much more text and no periods of silence in between, (b) are spoken by a different person than the meditation conditions, (c) using a louder volume, and (d) with references to tension/anxiety, which may have induced such emotions in listeners (e.g. in the excerpt of *Silverview*, the main character is experiencing anxiety). Therefore, any differences found may easily be due to one or more of these characteristics, rendering the results completely useless.

Authors' Response: We thank the reviewer for these comments, as they raise important issues that we would like to clarify. To address them, we decided to reanalyze our data to verify the robustness of our findings. We answer Reviewer 1's concerns point-per-point:

- (a) *The control conditions have much more text and no periods of silence in between:* We acknowledge that there are differences between the control and mindfulness conditions in terms of periods of silence, but not in the length of the segments. For the latter, we provide the audio spectrogram images for one of the mindfulness conditions (Body Scan) and one of the control conditions (the excerpt from '*Silverview*' by John Le Carre). It is immediately evident that the two recordings are matched in terms of length (15.01 minutes for '*Silverview*' vs. 15 minutes for Body Scan), as well as the frequency of the sounds of the two tracks, as shown by the sound pattern illustrated in the image below:

[REDACTED]

Regarding the periods of silence incorporated in the mindfulness exercises vs. the control conditions: guided mindfulness tracks deliberately incorporate pauses to enhance the effectiveness of meditation practices. Not having them would be unnatural and provide very limited ecological validity. Moreover, our exercises were recorded by an accredited mindfulness practitioner, who decided where such periods of silence were necessary. While introducing inappropriate pauses in a reading of a book excerpt (i.e., the control conditions) may indeed match the excerpt more to the mindfulness exercise, it may also result in adding suspense or boredom, with the potential for the rise of unwanted emotions (e.g., tension, arousal, etc.) or violations of expectations through disfluency.

b) *The control conditions are spoken by a different person than the meditation conditions:* We believe that the reviewer's remark was an unfortunate misunderstanding, and we would like to clarify that all audio tracks from the multi-site studio were recorded by the same voice, namely by Christoph Spiessens, a [BAMBA](https://www.christophspiessens.com/) Registered Mindfulness Teacher (MBSR): <https://www.christophspiessens.com/>. We have also included the information pertaining to the audio file recording in the Materials section (Page 9).

c) *The control conditions use a louder volume [than the experimental conditions]:* The volume at which each text was played was decided by every participant. We believe that the large sample involved in this study allows us to reasonably expect similar variations in volume across both experimental (i.e., mindfulness conditions) and control conditions. Although we tried to match the experimental and control conditions as much as possible, there are inherent differences stemming from the distinct nature of the experimental and control conditions, and it is challenging to completely eliminate them while maintaining the integrity of the study design. For example, we could not have reasonably asked each participant to play the audio file at a certain volume. Moreover, apart from the difference in the potential to induce increased anxiety by the 'Silverview' excerpt, we think that introducing at least some variability in theoretically irrelevant dimensions like length, default volume, or verbal density is needed to increase the external validity of the design. We agree that it may decrease the internal validity to some extent, which is always a trade-off experimenters have to make.

d) *The control conditions may have induced such emotions in listeners (e.g. in the excerpt of Silverview, the main character is experiencing anxiety):* We agree that the 'Silverview' excerpt might have potentially induced anxiety. However, as we attempted to control for some random variation due to a single control condition, we included three different control conditions. This allowed us to conduct additional sensitivity analyses. To respond to this concern analytically, we have added a dedicated section called "Robustness Analyses" in the updated version of the manuscript, at Page 5. Namely, we wrote the following:

We examined whether the difference in stress levels between the experimental (mindfulness) and the control conditions could be due to one particular story excerpt (i.e., 'Silverview' by John le Carré²¹) being perceived as more anxiogenic than the others (see Table 4). We conducted three independent *t*-tests and found a slight discrepancy in

self-reported stress levels between participants who listened to le Carré’s ‘Silverview’ excerpt and those who listened to Tolkien’s ‘Smith of Wootton Major’ excerpt²², $t(313.75) = 2.71$, $p = 0.007$. To address the issue of multiple comparisons, we applied the Bonferroni correction, yielding an adjusted p -value of 0.021. Notably, even after applying the Bonferroni correction, a statistically significant difference persisted between the two excerpts. These results indicate that participants exposed to the ‘Silverview’ excerpt experienced higher levels of stress compared to those exposed to the ‘Smith of Wootton Major’ excerpt. To test whether this may have affected the overall results, we re-ran the main analyses excluding participants who listened to ‘Silverview’ ($N = 157$). This participant exclusion resulted in a significant decrease in power, because the control group was reduced from 478 to 321 participants; nevertheless, the Bayes Factor remained above the threshold for compelling evidence in three of the mindfulness conditions (i.e., Body Scan, Mindful Breathing, Loving-Kindness), but not in the Mindful Walking condition ($BF_{10} = 0.08$, see Table 5).

Table 4. Means and Standard Deviations of scores on the State-Trait Anxiety Inventory, Form Y-1 in each control condition

Story excerpt	Length (minutes)	Word count	M	SD
‘Silverview’ by John le Carré	15.01	1838	2.04	0.51
‘The Old Man and the Sea’ by Ernest Hemingway	14.19	2039	1.93	0.47
‘Smith of Wootton Major’ by J. R. R. Tolkien	14.58	2309	1.88	0.50

Table 5. Results of the four independent comparisons with the active controls for the State-Trait Anxiety Inventory, Form Y-1, after excluding participants that listened to the ‘Silverview’ excerpt.

Condition	N	M	SD	BF ₁₀
Active Control Conditions	321	1.91	0.49	/
Body Scan	449	1.68	0.46	2.3 x 10 ⁵
Mindful Breathing	469	1.73	0.50	16.65
Loving-Kindness	427	1.70	0.49	118.10
Mindful Walking	416	1.73	0.46	0.08

Details about participants and Information Disclosure

- **Reviewer 1 Comment 4:** In addition, it is unclear how participants were recruited
- **Reviewer 3 Comment 3:** Where and how did the sites collect the participants? Through which channels?

Authors' Response: Participants were recruited by the 37 sites involved in the data collection process, which took place from March 23rd to June 30th, 2022. The specific channels through which participants were recruited were already outlined on Page 21 of the original manuscript. However, we added a clarification regarding the number of sites that participated in the data collection at Page 9 of the revised version:

Thirty-seven sites participated in the data collection (see the full list at <https://osf.io/uh3pk>). Participants could be recruited through the SONA system of the respective institution or via crowdsourcing platforms such as mTurk or Prolific. Participants could come from any geographic area if they met our inclusion criteria and could be given either credits or financial compensation in exchange for participating in the study.

- **Reviewer 1 Comment 5:** In the information letter, the aim of the study is disclosed (study on how participants would feel after meditation), which likely introduced bias. As a consequence, it was pointless for me reading the discussion.

Authors' Response: We regret that the disclosure we have made to participants, and its potential detrimental consequence, detracted Reviewer 1 from engaging with the discussion. We would like to take you through the process that we think legitimizes our decision to disclose the study purpose to participants. We have included some of the following arguments in the discussion of the study limitations (Pages 8 and 9):

- 1) We reasoned that we need to do that because in many fields of behavioral science it is indeed common practice to not deceive participants, for example because deception might affect participant retention as well as subsequent behavior (see, e.g., Jamison et al., 2006).
- 2) Because the research was led by a research group based in a British psychology department, we had to comply with the British Psychology Society's *Code of Human Research Ethics* that states the following:

“Deception or covert collection of data should only take place where it is essential to achieve the research results required, where the research objective has strong scientific merit and where there is an appropriate risk management and harm alleviation strategy” (page 24).

As such, we decided that our study needs participants to be fully informed that they might be allocated to a mindfulness condition because mindfulness practice – even short-term – is a form of psychological intervention and people need to be informed about what they might experience. Moreover, we know that mindfulness practice can be detrimental to some individuals (e.g., see Britton, W.B. (2019). *Can mindfulness be too much of a good thing? The value of a middle way.* *Current Opinions in Psychology: Special Issue on Mindfulness.*, 28, 159-165), and we needed to mitigate against this potential risk.

- 3) We believe that it would have been of limited use to deceive participants about the true nature of the study because those in the mindfulness condition might have easily guessed the purpose of the study based on the instructions alone.
- 4) We needed to inquire a priori whether participants were previous mindfulness practitioners or not, in order to exclude the former from participating. This was an essential exclusion criterion, and we could not have inquired about one's meditation experience post data collection because that would have wasted resources to collect data that then might have been discarded. Our previous meta-analysis showed no effect of brief self-administered mindfulness interventions, due to a variety of methodological shortcomings; in the absence of such a foundational effect, we believed that it was unwise to invest unlimited resources.

- **Reviewer 2 Comment 2:** Exclusion of Participants: Participants were excluded if they reported currently having or having had a history of mental illnesses. However, it might be important to specify what kind of mental illnesses were considered for exclusion.

- **Reviewer 3 Comment 4:** It is stated that “Participants were excluded if they reported currently having or having had a history of mental illnesses assessed via a pre-screening question..” How was this asked from the participants and what is included in mental illnesses here?

Authors’ Response: We appreciate the reviewers for bringing this concern to our attention. The survey was designed to terminate for participants who self-reported any diagnosis of a mental health illness. To assess participants' self-reported history of mental illnesses, we included a pre-screening question in the survey, namely: “Do you have - or have you had - a diagnosed, on-going mental health/illness/condition?” While this choice may have excluded more participants than necessary, we decided to screen out participants who reported such history because of growing evidence showing that mindfulness interventions could potentially result in psychotic episodes, panic attacks, depersonalization, or self-related changes (e.g., Van Gordon et al., 2017; Britton, Lindahl, Cooper, Canby & Palitsky, 2021; Goldberg, Lam, Britton & Davidson, 2021).

To provide further transparency and accessibility, we have made the complete survey transcript available on the OSF page of the project. The document titled "measurements multi-site" contains the survey questions in their entirety (<https://osf.io/c6nb8>).

- **Reviewer 3 Comment 5:** On page 13, it is reported that “We asked students to indicate in which university they were studying and asked non-students their current occupation” Does this mean that there were no other students but university students? Why?

Authors’ Response: Our participant pool indeed included university students and participants from the general population (i.e., non-students). Participation in our survey was also open to non-students who could be recruited via Prolific or from the sites that participated in the data collection. To clarify this issue, we revised the manuscript which now reads (Page 10):

“**Demographics.** Participants provided information regarding their age, gender, country of birth, country of residence, whether they were students or not, which university they were studying at (for the former), and what was their current occupation (for the latter).”

- **Reviewer 3 Comment 6:** The participants were not randomly selected. Thus, there may be bias that limits the generalizability of the findings. This should be discussed.

Authors' Response: We understand the Reviewer's point. Indeed, the majority of participants were students. While we are aware that this way of selecting participants limits the generalizability of the findings, we acknowledged that on Page 7 of the revised manuscript, as a constraint on generality:

Our findings only apply directly to participants who are 1) older than 18, 2) are fluent/native English speakers living in Australia, Europe, the UK, Canada, and the US, 3) are non-meditators, 4) do not have a history of mental illness and finally 5) are mostly students (94.2%). Further research is needed to test whether the findings of the present study will generalize to other populations.

Furthermore, causal inference in an experiment only assumes random allocation into groups in order to be valid (see Fisher, 1935). That is because causal inference is all about relative effectiveness and random sampling from population is orthogonal to that issue. Please note how practically all clinical RCTs in medicine study non-random sampled populations (patients). Please see <https://www.fharrell.com/post/rct-mimic/index.html> for a more elaborate short discussion of this and why this is a non-issue if the research question is about relative efficacy or effectiveness of treatments vs controls.

Analysis Plan

- **Reviewer 1 Comment 6:** The section on simulation of Bayesian is difficult to follow and seems unnecessary altogether (including Figure 1), except a brief statement. The Bayesian analysis is difficult to understand or follow and given the large sample size and simple model, a more conventional analysis would be preferred to most readers.
- **Reviewer 2, comment 1:** The manuscript presents a comprehensive multi-site study on self-administered mindfulness and its effects on stress regulation among English speakers. The study is well-structured, provides valuable insights, and showcases an excellent use of Bayesian statistics. There are only a few areas that could benefit from further clarification and discussion.

Authors' Response: While we fully understand the motivation behind the suggestion to remove the Bayesian analysis for simplicity's sake, we believe that retaining it would add to the

manuscript's scope and substance. The Bayesian approach we have adopted here is neither a self-serving, nor an optional statistical exercise. Instead, it is an integral, pre-registered component that serves as the backbone for both our participant sampling procedure and hypothesis testing. This methodology was not arbitrarily chosen, but justified in detail in the section "Simulation of the Bayesian design" (Page 12). In this section, we presented a simulation that demonstrated how the use of this adaptive design enhances informativeness and/or resource efficiency.

Specifically, we stated the following on Page 10 of the revised manuscript:

For the test of a single condition against controls, the sequential design is expected to be 27% more effective than collecting a fixed maximum N per lab, with the average N at the stopping point (BF boundary and maximum N) at .526. Even conservatively assuming a balanced- N situation, the informativeness of the design thus appeared to be adequate and the use of the adaptive design would likely enhance informativeness and/or resource efficiency."

The aim of the Bayesian approach was to enhance the informativeness and resource efficiency of our study. Thus, removing the Bayesian elements may be seen as compromising not just the integrity of study, but also its methodological rigor, as validated by our supporting simulations. Given these considerations, we propose retaining this integral aspect of our study. For readers unfamiliar with the Bayesian sequential sampling designs, we cite an introductory, accessible paper by Schönbrodt, F. D., Wagenmakers, E. J., Zehetleitner, M., & Perugini, M. (2017).

Sequential hypothesis testing with Bayes factors: Efficiently testing mean differences.

Psychological Methods, 22(2), 322–339.

- **Reviewer 2 Comment 3:** Moderation Analyses: The exploratory analyses investigated whether neuroticism and English level moderated the effects of mindfulness exercises on stress. However, the results of these analyses were inconclusive. It might be worth discussing potential reasons for these inconclusive results and how future research could further investigate these potential moderating effects.
- **Reviewer 3 Comment 8:** The moderation effects of neuroticism are examined by merging the different mindfulness conditions. However, there could be personality related differences in what kind of mindfulness training is most effective. It would be interesting to see the results for individual mindfulness tasks as well.

Authors' Response: Our analyses were inconclusive due to limited power to test moderation. We tried to increase power by merging the different mindfulness conditions and compared the merged conditions to the active control conditions, but still this was not enough to test for the moderation. If we were to test the moderation for each intervention separately, the statistical power allowed by our sample would be extremely small to allow us any meaningful conclusions. Consequently, the lack of a moderation effect for neuroticism observed in our study may be attributed to the insufficient power of our experimental design to detect such an effect or to a genuine absence of the effect; it is difficult to discern these two possibilities. While we agree that each intervention could produce differences, we simply do not have the possibility to study them with the current dataset. Future studies could explore the role of neuroticism in stress reduction by testing a single mindfulness exercise with a larger sample size to achieve greater statistical power, or test multiple interventions with the same participant to achieve greater control over individual differences. In our meta-analyses (Sparacio et al., 2023), we provided a protocol for how individual differences could be meta-analyzed over a large number of studies and suggestions to improve studies on self-administered mindfulness.

- **Reviewer 3 Comment 7:** On page 13, it is reported that the participants filled in the self-assessment Manikin scale. This scale is again referred to on page 14 under the heading Exploratory analysis. However, the analyses on this scale are not presented in the results nor discussed in the discussion section. It is unclear why this measure is mentioned at all.

Authors' Response: We thank the reviewer for raising this issue. We decided to exclude it from the main text because this measure was secondary to the main objective of the study. We have now included a separate document on the “Supplementary analyses” Open Science Framework (OSF) page, that includes a detailed presentation of the results obtained from the Self-Assessment Manikin scale, addressing the exploratory analysis mentioned earlier. The presentation is therefore provided together, in one document, and due to the tangential nature of the measure, provided in the supplemental materials.

Discussion and Conclusions

- **Reviewer 2 Comment 4:** I appreciate the authors' focus on brief mindfulness meditation in this study. In my own research, I have often encountered skepticism about the efficacy of brief meditation sessions in influencing cognitive and social cognitive outcomes. This study provides valuable evidence that even short periods of mindfulness practice can have a significant (but temporary) impact. I would encourage the authors to further emphasize this point in their discussion. The potency of brief meditation sessions, as

demonstrated in this study, could be a powerful counter-argument to the notion that longer sessions are always necessary. Highlighting this could not only strengthen the current study but also contribute to a broader understanding in the field about the potential of brief mindfulness interventions.

- **Reviewer 3 comment 9:** I would recommend adding more discussion on the relevance and meaning of the results. Why is this study important? What is the practical meaning of these results? Are there recommendations that the authors give based on their results and previous literature? What is the most important take home message of this study?

Authors' Response: We thank the reviewers for appreciating the focus of our study on brief mindfulness interventions. In our opinion, the practical significance of the findings is primarily methodological, as prior studies on the topic were not well-powered and were not pre-registered (Folk & Dunn 2023); the current study attempted to fill these gaps, and we improved the presentation of the core methodological meaning (now on Pages 6-7). Nevertheless, we agree with reviewers' suggestions that we can further emphasize how even short interventions can have a significant – albeit small – impact on short-term stress reduction or stress regulation. We have thus included a more substantial discussion of the meaning of these findings in the Discussion section (Page 6):

This project is an important step toward obtaining high-powered tests of the efficacy of self-administered mindfulness exercises for reducing stress. This multi-site study showcases how even short mindfulness exercises can be valuable tools in situations when short-term mood regulation is necessary, such as withstanding an exam, or calming oneself in a road rage situation²⁶. The possibility that short-term mindfulness practice adds to one's repertoire of skills to reduce stress need not harm nor challenge the established view that mindfulness meditation brings about positive results only via prologued practice. We argue that the current results align with previous evidence on strategies that help short-term mood repair without bringing about aversive consequences such as procrastination,²⁷ or other unhealthy coping strategies²⁸. By demonstrating that even short exercises can have a measurable impact, we hope to motivate individuals who may think that only engaging in lengthy mindfulness protocols is useful. Knowing that even brief mindfulness interventions can be beneficial for how people regulate their emotional responses to everyday stressors might increase their motivation to engage with these practices.

Longer mindfulness protocols like Mindfulness-Based Stress Reduction (MBSR) have been shown to enhance trait mindfulness, which refers to the dispositional ability to be present in the current moment^{15,29}. This trait mindfulness has been found to predict reduced reactivity to emotional suppression³⁰. Therefore, for individuals who already

possess high levels of trait mindfulness, the timing of mindfulness exercises may be less crucial, as they already exhibit a disposition that reduces susceptibility to stressors. However, for individuals who are new to mindfulness practice and may be encountering acute stress, an effective approach could involve engaging in even short mindfulness exercises to alleviate the tension that follows such stress.

- **Reviewer 3 Comment 10:** While discussing generalizability the authors use the word generality. Generalizability would seem more suitable a term.

Authors' Response: We appreciate the reviewer for bringing up this observation. We would like to note that the term "generality" was utilized in the original work by Simons et al. (2017). We retained their terminology to describe the limits of generalizability/generality.

- **Reviewer 3 Comment 11:** The authors mention (page 23, starting on line 518) that “These interventions should be intended as being effective in the short-term and are unlikely to affect dispositional traits (such as chronic stress). “It is no doubt true that a short intervention is unlikely to induce permanent changes at least if the mindfulness practice is not continued after the intervention. However, I would like to see the authors commenting on conditions in which mindfulness exercises could induce longer-term effects and what would be optimal timing for mindfulness practices. It has been suggested that long-term mindfulness practice that is started before the stress occurs, may be more effective in regulating stress, as compared to mindfulness practice that is started after the acute stress situation is already on. If one is new to mindfulness practice, it may be difficult to learn and concentrate on it, while simultaneously trying to cope with acute stress. A more effective approach could be to start practicing while the stress is not on and be prepared beforehand before the stress is encountered. This kind of discussion on optimal way of doing mindfulness practice would be important and give needed depth to the discussion. This would give a more comprehensive picture about the effectiveness of mindfulness in stress reduction.

Authors' Response: We thank the reviewer for raising an important point regarding the potential for longer-term effects of mindfulness exercises and the optimal timing for their practice. Given the specific design of our study, centered around a single, short-term mindfulness exercise, we refrain from speculating about an optimal timeframe for mindfulness practice. A follow-up study that systematically explores variations in timing would indeed be valuable. By comparing different timing scenarios, researchers can gain a clearer understanding of the potential benefits and drawbacks of various approaches. For instance, it would be possible to test whether concise

mindfulness interventions yield heightened stress resilience when administered prior to or subsequent to stress-inducing tasks, such as the Trier Social Stress Test (TSST; Kirschbaum et al., 1993). This approach could potentially uncover an optimal timing that maximizes the effectiveness of brief mindfulness interventions.

Other comments or suggestions:

- **Reviewer 1 Comment 7:** Abstract misses information on participants (at least age and sex), standard deviations of the reported means. Also $d = .56$ is not small, but medium.

Authors' Response: We appreciate the reviewer's comment. We have revised the abstract by incorporating the necessary information. We suspect that the reviewer bases themselves for effect size standards on Cohen's guidelines (with $d = 0.20$ or $r = 0.10$ interpreted as small effects, $d = 0.50$ or $r = 0.30$ as medium effects, and $d = 0.80$ or $r = 0.50$ as large effects). Hemphill (2003) noted that "it seems too simplistic to have a single set of empirical guidelines for interpreting the magnitude of correlation coefficients". We think it is more meaningful to interpret the change on the self-report instrument, which was 0.27 for the comparison between the condition with the largest reduction in stress levels (Body Scan; $M = 1.68$, $SD = 0.46$) and the control conditions ($M = 1.95$, $SD = 0.50$).

- **Reviewer 1 Comment 8:** Table 1 is not clear: what do the M (SD) represent ("comparison" is not specific enough), a Note is needed to explain BF_{10} . Unclear abbreviations (e.g., SONA).

Authors' Response: In Table 1, M represents the means, while SD represents the standard deviations of the experimental conditions and the active control conditions related to the main outcome of the study (i.e., self-reported stress). To enhance specificity, we have replaced the term "comparison" with "Bayesian mixed-effects models" in the table description, where we also detailed the meaning and significance of BF_{10} (Page 4 in the current manuscript, and Table 1 here included). By "SONA system" we refer to the full name of the platform used to recruit participants for universities (see <https://www.sona-systems.com/>), and we detailed its meaning in text (Page 9).

Table 1. Means and Standard deviations of self-reported stress levels of the four Bayesian mixed-effects models with the active controls for the State-Trait Anxiety Inventory, Form Y-1. A positive Bayes Factor (BF_{10}) denotes increasing evidence of H_1 compared to H_0 .

Condition	N	M	SD	BF ₁₀
Active Control Conditions	478	1.95	0.50	/
Body Scan	449	1.68	0.46	3.7 x 10 ¹¹
Mindful Breathing	469	1.73	0.50	2.3 x 10 ⁵
Loving-Kindness	427	1.70	0.49	1.1 x 10 ⁷
Mindful Walking	416	1.73	0.46	4.8 x 10 ²

- Reviewer 2 Comment 5:** I would like to draw the authors' attention to Figure 3 in the manuscript. While the figure provides important information, its current visual quality could be improved for better readability. Specifically, some of the numbers and site names are difficult to read due to overlapping elements. This overlap obscures important information and could potentially confuse readers. Additionally, the legend in the lower right corner could benefit from clearer and more consistent naming conventions. Lastly, the overall sharpness of the figure appears to be low, which further hampers readability. I would recommend revising this figure to enhance its clarity and effectiveness in conveying the intended information.

Authors' Response: We acknowledge the importance of visual quality and readability in conveying information effectively. Based on the reviewer's comments, we improved resolution to enhance the figure overall clarity in the newer version of the manuscript. We have also revised the naming conventions to ensure clarity and consistency, making it easier for readers to interpret the information provided. In the current form, the legend states the following:

“Fig. 1 | Forest plot (right side), plotted means and *SDs* for self-reported levels of stress respectively for the Body Scan and Mindful Breathing conditions compared to the active control conditions (left side)

Fig. 2 | Forest plot (right side), plotted means and *SDs* for self-reported levels of stress respectively for the Loving-Kindness and Mindful Walking conditions compared to the active control conditions (left side)”

- **Reviewer 3 Comment 12:** Please, remove meta-text from the discussion. This is such a short text that meta-text is not needed. (e.g., page 21 starting from line 455, page 22 starting from line 481) I would also recommend removing the heading “Strengths of the current study” since this seems to be a general discussion of the results, which is missing elsewhere.

Authors’ Response: We thank the reviewer for this comment. We have made the necessary revisions and removed the two sections of meta-text mentioned, as well as the heading "Strengths of the current study".

Decision Letter, first revision:

13th December 2023

Dear Dr Sparacio,

Thank you for your patience during the re-review of your manuscript. We have now heard from Reviewers 1 and 3 from the previous round of review. As you can see, although Reviewer 1 is fully satisfied with your revisions, Reviewer 3 has a number of outstanding comments that we ask that you address before we can make a final decision on publication. Additionally, we would like to clarify some key details about your study.

As you may be aware, Nature Human Behaviour has specific requirements regarding the registration of clinical trials (please see our (pre)registration policy: <https://www.nature.com/nathumbehav/editorial-policies/preregistration-policy>).

We classify as clinical trials studies that meet the WHO definition:

"For the purposes of registration, a clinical trial is any research study that prospectively assigns

human participants or groups of humans to one or more health-related interventions to evaluate the effects on health outcomes. Clinical trials may also be referred to as interventional trials. Interventions include but are not restricted to drugs, cells and other biological products, surgical procedures, radiologic procedures, devices, behavioural treatments, process-of-care changes, preventive care, etc."

Your study meets the definition of a clinical trial (your intervention is behavioural and the outcome measure - stress - is health-related). For clinical trials, our requirement is for prospective registration in a WHO-approved clinical trial registry (e.g., clinicaltrials.gov) prior to the enrollment of the first participant. We appreciate that this practice, although well-known and established among biomedical scientists, is less known among social and behavioural scientists. For this reason, we also accept clinical trials carried out by social and behavioural scientists that were not prospectively registered in a clinical trial registry, but instead in a general registry, such as OSF, provided that preregistration occurred prior to the enrollment of the first participant.

In examining your preregistration, we identified that, as noted in the manuscript, preregistration took place on March 22, 2022. However, your OSF materials include a document called "Data_Collection_Tracker.Rmd", which was first created in June 2021 (and then updated several times). We also noted that in your March 22 preregistration the analysis plan is written in the past tense.

With your revised manuscript, please confirm whether data collection started ahead of the preregistration of the project. To support your response, please include time-stamped evidence that clearly demonstrates precedence of the preregistration over the onset of data collection.

Additionally, in order to comply with our policy for (pre)registration, please address the following points:

1. You preregistered exploratory analyses examining the effects of any of the experimental conditions as compared to the active control conditions on the dimensions of pleasure, arousal and dominance. They are mentioned in the paper, but it seems that the results of those analyses are not reported in the paper.
2. It appears that the exploratory test for moderation effects of English language proficiency was not preregistered. Please state this deviation from the preregistration in your manuscript.
3. You preregistered one control condition and mentioned that three different stories would be used, in the paper you refer to "three active control conditions". Please explicitly clarify this deviation in your revised manuscript.
4. We ask that you transparently report all deviations from your preregistration in your manuscript.

In sum, we invite you to revise your manuscript taking into account all reviewer and editor comments. We are committed to providing a fair and constructive peer-review process. Do not hesitate to contact us if there are specific requests from the reviewers that you believe are technically impossible or unlikely to yield a meaningful outcome.

We hope to receive your revised manuscript within 4-8 weeks. I would be grateful if you could contact us as soon as possible if you foresee difficulties with meeting this target resubmission date.

- Include a "Response to the editors and reviewers" document detailing, point-by-point, how you addressed each editor and referee comment. If no action was taken to address a point, you must provide a compelling argument. This response will be used by the editors and reviewers to evaluate your revision.
- Highlight all changes made to your manuscript or provide us with a version that tracks changes.

[REDACTED]

We look forward to seeing the revised manuscript and thank you for the opportunity to review your work. Please do not hesitate to contact me if you have any questions or would like to discuss these revisions further.

Sincerely,

[REDACTED]

REVIEWER COMMENTS:

Reviewer #1:

Remarks to the Author:

The comments have been dealt with thoroughly and satisfactorily.

Only in the abstract the dependent variable should be made specific ("stress" is unclear).

Reviewer #2:

None

Reviewer #3:

Remarks to the Author:

For the most part the authors have responded to my comments sufficiently. However, some concerns do remain.

For your next response letter, please respond individually to each question without combining them with responses to other reviewer's comments, as that makes it rather difficult to go through your responses.

1. The following comment was not clarified sufficiently: "On page 13, it is reported that "We asked students to indicate in which university they were studying and asked non-students their current occupation" Does this mean that there were no other students but university students? Why?"

The authors collected data from general population. Did that data query whether the participant was a student? Did that data not include any students other than university students?

2. My previous comment: "While discussing generalizability the authors use the word generality. Generalizability would seem more suitable a term."

The authors did not change their manuscript. Instead the authors give a single reference where the term generality has also been used. This is not a sufficient justification.

3. The following comment was also ignored by the authors: "The authors mention (page 23, starting on line 518) that "These interventions should be intended as being effective in the short-term and are unlikely to affect dispositional traits (such as chronic stress). " It is no doubt true that a short intervention is unlikely to induce permanent changes at least if the mindfulness practice is not continued after the intervention. I would like to see the authors commenting on conditions in which mindfulness exercises could induce longer-term effects and what would be optimal timing for mindfulness practices. It has been suggested that long-term mindfulness practice that is started before the stress occurs, may be more effective in regulating stress, as compared to mindfulness practice that is started after the acute stress situation is already on. If one is new to mindfulness practice, it may be difficult to learn and concentrate on it, while simultaneously trying to cope with acute stress. A more effective approach could be to start practicing while the stress is not on and be prepared beforehand before the stress is encountered. This kind of discussion on optimal way of doing mindfulness practice would be important and give needed depth to the discussion. This would give a more comprehensive picture about the effectiveness of mindfulness in stress reduction."

There are several ways in which the authors could have taken this comment into account in their manuscript but they chose not to, justifying this choice by stating that they refrain from speculating. The comment did not request speculation. What it does request is a research-based discussion. I'm surprised if there is nothing at all (without speculation) that the authors can say about the points that were risen.

Author Rebuttal, first revision:***Responses to Reviewers' comments******Reviewer 1***

Comment 1: The comments have been dealt with thoroughly and satisfactorily. Only in the abstract the dependent variable should be made specific ("stress" is unclear).

Response: We thank the reviewer for these very kind words! We rephrased the sentence in the abstract and made the variable characterization more specific. The abstract now reads:

"Our findings suggest that mindfulness may be beneficial for reducing self-reported short-term stress for English speakers from higher-income countries."

Reviewer 3

Remarks to the author: For the most part the authors have responded to my comments sufficiently. However, some concerns do remain. For your next response letter, please respond individually to each question without combining them with responses to other reviewer's comments, as that makes it rather difficult to go through your responses.

Response: We thank the reviewer for this feedback. In the present response letter, we committed to address each of the concerns individually.

Comment 1: The following comment was not clarified sufficiently: "On page 13, it is reported that "We asked students to indicate in which university they were studying and asked non-students their current occupation" Does this mean that there were no other students but university students? Why? The authors collected data from the general population. Did that data query whether the participant was a student? Did that data not include any students other than university students?"

Response: We thank the reviewer for bringing attention to this aspect, and we appreciate the opportunity to provide further clarification. Participants were indeed queried about their student status; if they identified as students, they were further asked to specify their university affiliation. It is important to note that the inclusion criteria did not exclude students who attended other

institutes than universities (e.g., trade schools or community colleges). While our survey captured information on whether participants were students or not, it does not differentiate between university students and those enrolled in other educational institutions. Nevertheless, we believe that the proportion of non-university students included in our sample is minimal, because the recruitment channels primarily utilized institutional systems such as the SONA system (i.e., Participant Pool Management for universities), typically used in higher education/university settings alongside possibly other institutes, such as trade schools, and because our inclusion criterion of participants being adults, namely 18 years or older.

Comment 2: My previous comment: "While discussing generalizability the authors use the word generality. Generalizability would seem more suitable a term." The authors did not change their manuscript. Instead the authors give a single reference where the term generality has also been used. This is not a sufficient justification.

Response: We understand the reviewer's concern. We had previously retained the term 'generality' based on Simons et al. (2017) that advocates for a 'Constraints on Generality' (COG) statement in the discussion section to justify target populations for the reported findings. In response to the reviewer's suggestion, we have refrained from using their term, and revised the statement using the term 'generality' and used 'generalizability' instead, to increase clarity and accessibility for readers not familiar with the Simons et al.'s terminology.

In the revised manuscript on Page 7, the text now reads:

“Finally, we believe that it is important to consider several limitations on the generalizability of the results of this study.”

Comment 3: The following comment was also ignored by the authors: "The authors mention (page 23, starting on line 518) that “These interventions should be intended as being effective in the short-term and are unlikely to affect dispositional traits (such as chronic stress). “ It is no doubt true that a short intervention is unlikely to induce permanent changes at least if the mindfulness practice is not continued after the intervention. I would like to see the authors commenting on conditions in which mindfulness exercises could induce longer-term effects and what would be optimal timing for mindfulness practices. It has been suggested that long-term mindfulness practice that is started before the stress occurs, may be more effective in regulating stress, as compared to mindfulness practice that is started after the acute stress situation is already on. If one is new to mindfulness practice, it may be difficult to learn and concentrate on it, while simultaneously trying to cope with acute stress. A more effective approach could be to start practicing while the stress is not on and be prepared beforehand before the stress is encountered. This kind of discussion on optimal way of doing mindfulness practice would be

important and give needed depth to the discussion. This would give a more comprehensive picture about the effectiveness of mindfulness in stress reduction."There are several ways in which the authors could have taken this comment into account in their manuscript but they chose not to, justifying this choice by stating that they refrain from speculating. The comment did not request speculation. What it does request is a research-based discussion. I'm surprised if there is nothing at all (without speculation) that the authors can say about the points that were risen.

Response: We thank the reviewer for this comment and for allowing us another opportunity to integrate these points into a more elaborate, research-based discussion. We would also like to apologize for not providing this discussion earlier in the revision process. We agree that these questions are important for situating our multi-study results in the wider context of this topic. The current multi-site project does not directly address Reviewer 3's question about the 'optimal way' to practice mindfulness because this question is beyond the scope of our research. Nevertheless, we recognize the importance of this point, and have provided a more detailed discussion on Pages 6-7, where the amended text reads the following:

This project is an important step toward obtaining high-powered tests of the efficacy of self-administered mindfulness exercises for reducing stress. On the one hand, the current multi-site study showcases how even short mindfulness exercises can be valuable tools in situations when short-term mood regulation is necessary, such as withstanding a stressful exam, or calming oneself in a road-rage situation²⁶. The possibility that short-term mindfulness practice adds to one's repertoire of skills to reduce stress need not harm nor challenge the popular expectation that mindfulness meditation brings about positive results only via prolonged practice. Learning to practice mindfulness in a shorter time than traditional protocols typically require is a valuable asset for people for whom longer time commitment for mindfulness is a capacity- or a motivation-based deterrent²⁷. Understanding the optional timing to learn mindfulness skills or the conditions in which mindfulness induces effects that are longer-term compared to those observed in the current experiment are important questions, yet they extend beyond the scope of the current research. Notwithstanding the absence of high-powered, pre-registered studies that would make for a more reliable body of knowledge on these topics, some existent data yet allows partial answers. In line with the Extended Model of Emotion Regulation²⁸, mindfulness skills mastered before a stressful situation occurs can allow one extra flexibility to regulate antecedents of emotional reactions, such as which aspects one pays attention to ('attentional deployment') or the way one cognitively represents the stressful situation ('cognitive change'). For example, an 8-week randomized-controlled

trial of mindfulness completed in the year leading to the examination period significantly reduced students' psychological distress during that same examination period²⁹. Longer, e.g., 8-week mindfulness protocols such as Mindfulness-Based Stress Reduction (MBSR), can enhance trait/dispositional mindfulness (i.e., the inherent capacity to be in the present moment^{15,30}) and people's mindfulness self-efficacy (i.e., one's ability to maintain non-judgmental awareness in different situations). Therefore, for individuals who already possess high levels of trait mindfulness, the timing of mindfulness exercises may be less crucial, as they already exhibit a disposition that helps reduce their susceptibility to stressors. Nevertheless, additional pre-registered, high-powered studies need to be conducted on the topic to conclusively determine the ideal timing for mindfulness exercises and their potential for long-term changes.

Decision Letter, second revision:

23rd February 2024

Dear Dr Sparacio,

RE: "Self-administered mindfulness interventions reduce stress in a large, randomized controlled multi-site study"

Thank you for submitting your revised manuscript and for all your work on the revision.

Although your manuscript has been revised in response to reviewer comments, it does not fully comply with our editorial policies and formatting requirements. Specifically, please retroactively register your work on clinicaltrials.gov and explain the history of the manuscript. Please report the clinicaltrials.gov registration in the revised manuscript. My sincere apologies that our instructions were unclear in the previous decision letter. No additional changes are required at this stage.

[REDACTED]

Thank you in advance for attending to these requests and I look forward to receiving your revised manuscript.

Sincerely,

[REDACTED]

Author Rebuttal, second revision:***Responses to Reviewers' comments******Reviewer 1***

Comment 1: The comments have been dealt with thoroughly and satisfactorily. Only in the abstract the dependent variable should be made specific ("stress" is unclear).

Response: We thank the reviewer for these very kind words! We rephrased the sentence in the abstract and made the variable characterization more specific. The abstract now reads:

"Our findings suggest that mindfulness may be beneficial for reducing self-reported short-term stress for English speakers from higher-income countries."

Reviewer 3

Remarks to the author: For the most part the authors have responded to my comments sufficiently. However, some concerns do remain. For your next response letter, please respond individually to each question without combining them with responses to other reviewer's comments, as that makes it rather difficult to go through your responses.

Response: We thank the reviewer for this feedback. In the present response letter, we committed to address each of the concerns individually.

Comment 1: The following comment was not clarified sufficiently: "On page 13, it is reported that "We asked students to indicate in which university they were studying and asked non-students their current occupation" Does this mean that there were no other students but university students? Why? The authors collected data from the general population. Did that data query whether the participant was a student? Did that data not include any students other than university students?"

Response: We thank the reviewer for bringing attention to this aspect, and we appreciate the opportunity to provide further clarification. Participants were indeed queried about their student status; if they identified as students, they were further asked to specify their university affiliation. It is important to note that the inclusion criteria did not exclude students who attended other

institutes than universities (e.g., trade schools or community colleges). While our survey captured information on whether participants were students or not, it does not differentiate between university students and those enrolled in other educational institutions. Nevertheless, we believe that the proportion of non-university students included in our sample is minimal, because the recruitment channels primarily utilized institutional systems such as the SONA system (i.e., Participant Pool Management for universities), typically used in higher education/university settings alongside possibly other institutes, such as trade schools, and because our inclusion criterion of participants being adults, namely 18 years or older.

Comment 2: My previous comment: "While discussing generalizability the authors use the word generality. Generalizability would seem more suitable a term." The authors did not change their manuscript. Instead the authors give a single reference where the term generality has also been used. This is not a sufficient justification.

Response: We understand the reviewer's concern. We had previously retained the term 'generality' based on Simons et al. (2017) that advocates for a 'Constraints on Generality' (COG) statement in the discussion section to justify target populations for the reported findings. In response to the reviewer's suggestion, we have refrained from using their term, and revised the statement using the term 'generality' and used 'generalizability' instead, to increase clarity and accessibility for readers not familiar with the Simons et al.'s terminology.

In the revised manuscript on Page 7, the text now reads:

“Finally, we believe that it is important to consider several limitations on the generalizability of the results of this study.”

Comment 3: The following comment was also ignored by the authors: "The authors mention (page 23, starting on line 518) that “These interventions should be intended as being effective in the short-term and are unlikely to affect dispositional traits (such as chronic stress). “ It is no doubt true that a short intervention is unlikely to induce permanent changes at least if the mindfulness practice is not continued after the intervention. I would like to see the authors commenting on conditions in which mindfulness exercises could induce longer-term effects and what would be optimal timing for mindfulness practices. It has been suggested that long-term mindfulness practice that is started before the stress occurs, may be more effective in regulating stress, as compared to mindfulness practice that is started after the acute stress situation is already on. If one is new to mindfulness practice, it may be difficult to learn and concentrate on it, while simultaneously trying to cope with acute stress. A more effective approach could be to start practicing while the stress is not on and be prepared beforehand before the stress is encountered. This kind of discussion on optimal way of doing mindfulness practice would be

important and give needed depth to the discussion. This would give a more comprehensive picture about the effectiveness of mindfulness in stress reduction."There are several ways in which the authors could have taken this comment into account in their manuscript but they chose not to, justifying this choice by stating that they refrain from speculating. The comment did not request speculation. What it does request is a research-based discussion. I'm surprised if there is nothing at all (without speculation) that the authors can say about the points that were risen.

Response: We thank the reviewer for this comment and for allowing us another opportunity to integrate these points into a more elaborate, research-based discussion. We would also like to apologize for not providing this discussion earlier in the revision process. We agree that these questions are important for situating our multi-study results in the wider context of this topic. The current multi-site project does not directly address Reviewer 3's question about the 'optimal way' to practice mindfulness because this question is beyond the scope of our research. Nevertheless, we recognize the importance of this point, and have provided a more detailed discussion on Pages 6-7, where the amended text reads the following:

This project is an important step toward obtaining high-powered tests of the efficacy of self-administered mindfulness exercises for reducing stress. On the one hand, the current multi-site study showcases how even short mindfulness exercises can be valuable tools in situations when short-term mood regulation is necessary, such as withstanding a stressful exam, or calming oneself in a road-rage situation²⁶. The possibility that short-term mindfulness practice adds to one's repertoire of skills to reduce stress need not harm nor challenge the popular expectation that mindfulness meditation brings about positive results only via prolonged practice. Learning to practice mindfulness in a shorter time than traditional protocols typically require is a valuable asset for people for whom longer time commitment for mindfulness is a capacity- or a motivation-based deterrent²⁷. Understanding the optional timing to learn mindfulness skills or the conditions in which mindfulness induces effects that are longer-term compared to those observed in the current experiment are important questions, yet they extend beyond the scope of the current research. Notwithstanding the absence of high-powered, pre-registered studies that would make for a more reliable body of knowledge on these topics, some existent data yet allows partial answers. In line with the Extended Model of Emotion Regulation²⁸, mindfulness skills mastered before a stressful situation occurs can allow one extra flexibility to regulate antecedents of emotional reactions, such as which aspects one pays attention to ('attentional deployment') or the way one cognitively represents the stressful situation ('cognitive change'). For example, an 8-week randomized-controlled

trial of mindfulness completed in the year leading to the examination period significantly reduced students' psychological distress during that same examination period²⁹. Longer, e.g., 8-week mindfulness protocols such as Mindfulness-Based Stress Reduction (MBSR), can enhance trait/dispositional mindfulness (i.e., the inherent capacity to be in the present moment^{15,30}) and people's mindfulness self-efficacy (i.e., one's ability to maintain non-judgmental awareness in different situations). Therefore, for individuals who already possess high levels of trait mindfulness, the timing of mindfulness exercises may be less crucial, as they already exhibit a disposition that helps reduce their susceptibility to stressors. Nevertheless, additional pre-registered, high-powered studies need to be conducted on the topic to conclusively determine the ideal timing for mindfulness exercises and their potential for long-term changes.

Decision Letter, third revision:

5th April 2024

Dear Dr. Sparacio,

Thank you for your patience as we've prepared the guidelines for final submission of your Nature Human Behaviour manuscript, "Self-administered mindfulness interventions reduce stress in a large, randomized controlled multi-site study" (NATHUMBEHAV-23041191C). Please carefully follow the step-by-step instructions provided in the attached file, and add a response in each row of the table to indicate the changes that you have made. Please also address the additional marked-up edits we have proposed within the reporting summary. Ensuring that each point is addressed will help to ensure that your revised manuscript can be swiftly handed over to our production team.

We would hope to receive your revised paper, with all of the requested files and forms within two-three weeks. Please get in contact with us if you anticipate delays.

Nature Human Behaviour offers a Transparent Peer Review option for new original research manuscripts submitted after December 1st, 2019. As part of this initiative, we encourage our authors to support increased transparency into the peer review process by agreeing to have the reviewer comments, author rebuttal letters, and editorial decision letters published as a Supplementary item.

When you submit your final files please clearly state in your cover letter whether or not you would like to participate in this initiative. Please note that failure to state your preference will result in delays in accepting your manuscript for publication.

In recognition of the time and expertise our reviewers provide to Nature Human Behaviour's editorial process, we would like to formally acknowledge their contribution to the external peer review of your manuscript entitled "Self-administered mindfulness interventions reduce stress in a large, randomized controlled multi-site study". For those reviewers who give their assent, we will be publishing their names alongside the published article.

Cover suggestions

We welcome submissions of artwork for consideration for our cover. For more information, please see our guide for cover artwork.

ORCID

Non-corresponding authors do not have to link their ORCIDs but are encouraged to do so. Please note that it will not be possible to add/modify ORCIDs at proof. Thus, please let your co-authors know that if they wish to have their ORCID added to the paper they must follow the procedure described in the following link prior to acceptance:

Nature Human Behaviour has now transitioned to a unified Rights Collection system which will allow our Author Services team to quickly and easily collect the rights and permissions required to publish your work. Approximately 10 days after your paper is formally accepted, you will receive an email in providing you with a link to complete the grant of rights. If your paper is eligible for Open Access, our Author Services team will also be in touch regarding any additional information that may be required to arrange payment for your article.

Please note that *Nature Human Behaviour* is a Transformative Journal (TJ). Authors may publish their research with us through the traditional subscription access route or make their paper immediately open access through payment of an article-processing charge (APC). Authors will not be required to make a final decision about access to their article until it has been accepted. Find out more about Transformative Journals

Authors may need to take specific actions to achieve compliance with funder and institutional open access mandates. If your research is supported by a funder that requires immediate open access (e.g. according to Plan S principles) then you should select the gold OA route, and we will direct you to the compliant route where possible. For authors selecting the subscription

publication route, the journal's standard licensing terms will need to be accepted, including self-archiving policies. Those licensing terms will supersede any other terms that the author or any third party may assert apply to any version of the manuscript.

[REDACTED]

Best regards,
[REDACTED]

On behalf of

[REDACTED]

Final Decision Letter:

Dear Dr Sparacio,

We are pleased to inform you that your Article "Self-administered mindfulness interventions reduce stress in a large, randomized controlled multi-site study", has now been accepted for publication in Nature Human Behaviour.

Please note that *Nature Human Behaviour* is a Transformative Journal (TJ). Authors may publish their research with us through the traditional subscription access route or make their paper immediately open access through payment of an article-processing charge (APC). Authors will not be required to make a final decision about access to their article until it has been accepted. Find out more about Transformative Journals

Authors may need to take specific actions to achieve compliance with funder and institutional open access mandates. If your research is supported by a funder that requires immediate open access (e.g. according to Plan S principles) then you should select the gold OA route, and we will direct you to the compliant route where possible. For authors selecting the subscription publication route, the journal's standard licensing terms will need to be accepted, including self-archiving policies. Those licensing terms

will supersede any other terms that the author or any third party may assert apply to any version of the manuscript.

With best regards,

[REDACTED]